# FORLA: Federated Object-Centric Representation Learning with Slot Attention

**Guiqiu Liao**[1]    **Matjaz Jogan**[1]    **Eric Eaton**[2]    **Daniel A. Hashimoto**[1,2]

[1]PCASO Laboratory, Dept. of Surgery, University of Pennsylvania
[2]Dept. of Computer and Information Science, University of Pennsylvania
guiqiu.liao@pennmedicine.upenn.edu

## Abstract

Learning efficient visual representations across heterogeneous unlabeled datasets remains a central challenge in federated learning. Effective federated representations require features that are jointly informative across clients while disentangling client-specific factors without supervision. We thus introduce FORLA, a novel framework for federated object-centric representation learning and feature adaptation using unsupervised slot attention. At the core of our method is a shared feature adapter, trained collaboratively across clients to adapt features from foundation models, and a shared slot attention module that learns to reconstruct the adapted features. To optimize this adapter, we design a two-branch student–teacher architecture. In each client, a student decoder learns to reconstruct full features from foundation models, while a teacher decoder reconstructs their adapted, low-dimensional counterpart. The shared slot attention module bridges cross-domain learning by aligning object-level representations across clients. Experiments in multiple real-world datasets show that our framework not only outperforms centralized baselines on object discovery but also learns a compact, universal representation that generalizes well across domains. This work highlights federated slot attention as an effective tool for scalable, unsupervised visual representation learning from cross-domain data with distributed concepts. Our code, data, and pretrained models are available at: https://github.com/PCASOlab/FORLA.

## 1   Introduction

Self-supervised vision models such as DINO [7] and MAE [20] have popularized representation learning as a transferable pretraining strategy for diverse downstream tasks. However, their joint use in federated learning (FL) remains underexplored. Consider a FL scenario where clients hold data from different domains and sub-domains: for example, some contain expert domains like various types of surgical procedures, while datasets on other clients represent natural domains with everyday objects. Despite their differences, these datasets may share certain properties, such as being captured with visible light cameras, leading to overlapping textures or visual patterns. In some cases, specific objects, such as tissues and surgical instruments, are shared only within sub-domains of the expert domain, in this case across different surgical types. This heterogeneity poses challenges for FL using unsupervised methods that leverage consistency to capture semantic meaning in the feature space [7, 20, 30].

We propose that self-supervised object-centric learning [17, 38] offers the right inductive bias for learning semantically meaningful representations from independently curated datasets while preserving object-level knowledge across clients in federated settings. Slot Attention [38] learns object-centric representations by factoring a scene into independent *slots* that specialize in individual objects, yielding an interpretable, potentially domain-invariant basis. However, training slot attention

39th Conference on Neural Information Processing Systems (NeurIPS 2025).

on pooled data across clients entangles concepts and inflates computation, while training one model per client preserves local structure but leaves feature spaces incompatible.

We thus introduce **FORLA** (**F**ederated **O**bject-Centric **R**epresentation **L**earning with Slot **A**ttention). that couples object-centric structure learning with collaborative feature adaptation across clients. Each client starts from frozen foundation-model embeddings. A shared, lightweight adapter maps these embeddings into a compact latent space. We train the adapter through a two-branch teacher–student objective: (i) a teacher decoder reconstructs the learnable feature representation; (ii) a student decoder reconstructs the frozen foundation model representation; (iii) both teacher and student share the adapter and slot attention modules with other clients. Importantly, FL lets the student inherit the teacher's knowledge without a knowledge distillation (KD) loss, sidestepping the permutation-matching problem posed by slot attention's permutation invariance.

The dual-branch architecture of FORLA, with one branch reconstructing raw features while the other reconstructs dynamic features, is crucial for efficient learning. On the one hand, reconstructing raw features anchors the learning objective and enables convergence. On the other hand, learning adaptive features improves object representation in complex scenes and novel domains, as shown by the PCA visualization of adapted versus raw features in Fig. 2.

Object-centric learning has not been scaled to large multi-domain datasets, including mixtures of expert and natural domains. This work, to our knowledge, is the first to study and demonstrate the advantage of federated unsupervised object centric learning at scale. Our key contributions are: 1) We consider representation learning a first-class problem in FL and motivate object-centric learning as a principled solution. 2) We design an architecture with a communication-efficient teacher–student adapter and slot attention modules that harmonizes heterogeneous raw foundation features without sharing data. 3) Through extensive experiments we show that this architecture consistently outperforms centralized training on mixed data, even when using identical foundation models. For instance, federated slot attention achieves a 7.6% higher mIoU in multi-domain object discovery while reducing client–server communication by 85.2% through lightweight adaptation layers, compared to standard fine-tuning of foundation models. Furthermore, relative to the mixture-of-foundation-models approach, the communication cost can be reduced by up to $6.7\times$. This highlights FL's unique ability to exploit distributed datasets when using an object-centric approach.

## 2 Related work

In this section we provide a brief overview of related work for FORLA which we further expand in Supplementary Material A.

**Federated Learning (FL)**   While FL traditionally focuses on parameter aggregation [42], recent work addresses non-IID challenges through prototype exchange [44], contrastive alignment [69], and knowledge distillation [19, 63]. However, these methods focus on classification/segmentation rather than learning disentangled representations. Our work bridges this gap by introducing self-supervised object-centric learning to FL.

**Object-Centric Learning**   Slot Attention [38] introduced object-centric learning via iterative slot refinement. Follow-up work improved scalability [56], applied foundation models [52], and enhanced performance via feature augmentation [39], multimodal inputs [67, 2], or fine-tuning [12]. However, no prior work integrates slot attention with federated learning across heterogeneous data which is a setting where FL can mitigate concept entanglement by preserving local structure.

**Concept Entanglement in Multi-Domain Learning**   Concept entanglement persists as a key challenge in multi-domain learning with overlapping semantics. Early solutions like Domain Separation Networks [4] explicitly partition features into shared and private subspaces. Subsequent advances include adversarial domain-invariant encoding [37], variational information maximization [24], and label-free latent disentanglement via sparse adapters [9]. Recent work further distills domain-specific representations through adapter alignment [32]. While these methods reduce entanglement, none address object-centric disentanglement – a gap we bridge in both centralized and federated settings.

**Adaptation Layers and Feature Harmonization**   Adaptation layers align feature spaces in transfer [49] and multi-task learning [59]. In FL, adapters support personalization [8]. FT-DINOSAUR [12]

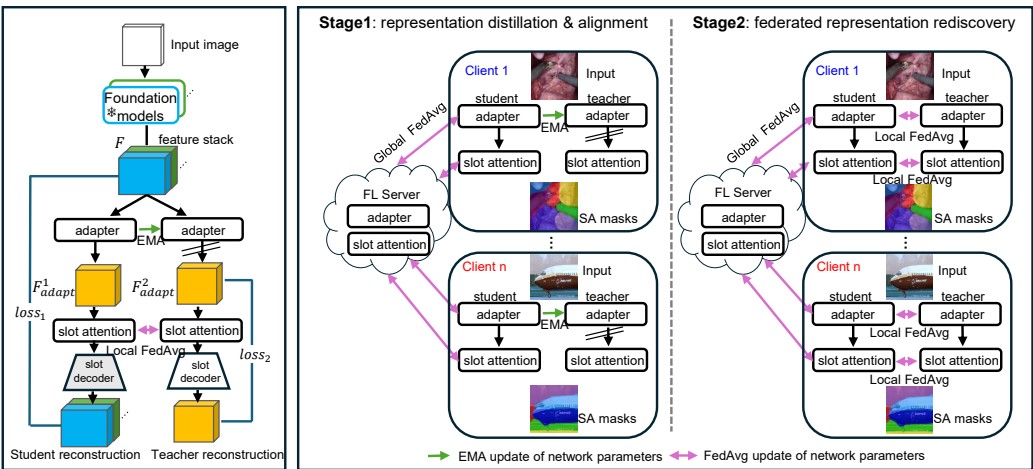

Figure 1: Overview of FORLA. Left: Within each client, student and teacher branches are trained to reconstruct raw foundation model features and adapted features, respectively. Right: During each global federated learning (FL) round, the student's adapter and Slot Attention (SA) modules are aggregated across clients via the server. In later stages of training, the teacher's adapter and SA modules are locally synchronized with the student through a local FedAvg update, enabling progressive knowledge distillation.

shows that object-centric learning benefits from cross-domain fine-tuning. Lightweight adapters [16, 70], bias tuning [6, 68], and low-rank updates [22] offer scalable adaptation. Masking-based adaptation for frozen models [60] enables harmonized multi-model input without the need to tune one particular foundation model. Our approach integrates these ideas in a federated setting to align local representations across diverse clients.

**Knowledge distillation**  Knowledge distillation has been used in federated learning in a fully supervised setting [71, 61]. Unsupervised self-distillation, as in DINO [7], ensembles knowledge via Exponential Moving Average (EMA) updates from student to teacher, akin to Polyak averaging [45]. We adopt a similar strategy: the teacher is updated by averaging with the student, without a conventional KD loss; a step necessary due to Slot Attention's permutation invariance [38]. Our method is also related to co-distillation [1], where student and teacher models mutually distill knowledge to perform efficient ensemble learning from large-scale, distributed data. Unlike standard co-distillation, which employs identical models, we share only a subset of modules between student and teacher, and our framework operates entirely in an unsupervised setting.

# 3 Method

## 3.1 Overview

Figure 1 illustrates the proposed framework for federated object-centric representation learning and feature adaptation using Slot Attention. Clients share frozen features extracted from multiple vision foundation models. On top of these shared features, FORLA trains two parallel branches with identical feature-adapter and Slot Attention architectures but with slot decoders of different sizes. Lightweight adapters compress and harmonize the stacked foundation features into a compact, low-dimensional representation. The two Slot Attention modules extract object-centric latent slots using the adapted features. The student decoder learns to reconstruct the raw high-dimensional frozen features, while the teacher decoder reconstructs the adapted features. During training, the adapter and Slot Attention parameters are periodically aggregated and synchronized across clients and within each clients' branches using FedAvg. This architecture enables FORLA to: (i) learn a global distilled representation efficiently, and (ii) align slot attention across heterogeneous domains without supervision.

### 3.2 Slot Attention with Adapted Foundation Features

**Feature stacking and adaptation**   Given an input image, we extract features from $M$ frozen foundation models (DINO, SAM [28], MAE, CLIP [48]). Let $F^{(m)} \in \mathbb{R}^{H \times W \times C_m}$ be the feature map from model $m$. We concatenate these into:

$$\mathbf{F} = [\, F^{(1)} \parallel F^{(2)} \parallel \cdots \parallel F^{(M)} \,] \in \mathbb{R}^{H \times W \times C_{\text{tot}}}, \quad C_{\text{tot}} = \sum_m C_m \tag{1}$$

To reduce dimensionality and fuse the stacked features, we introduce an *adapter module* $g_\phi$ producing adapted features $\mathbf{F}_{\text{adapt}} \in \mathbb{R}^{H \times W \times d}$:

$$\mathbf{F}_{\text{adapt}} = g_\phi(\mathbf{F}), \quad d \ll C_{\text{tot}} \tag{2}$$

We explore three feature adapter designs: *MLP adapter* [50], *mixture-of-experts (MoE)* [53, 15], and *Attention-based Feature Modulation (AFM)* [23]. Details are provided in Supplementary Material B.

**Slot attention**   The adapted features $\mathbf{F}_{\text{adapt}}$ are flattened into $N = H \times W$ vectors and fed into a Slot Attention encoder [38], producing $K$ object-centric slot latent vectors $\{s_k\}_{k=1}^{K}$. As in DINOSAUR [52], decoding reconstructs target features and slot attention masks using MLP decoders. Details on Slot Attention and decoding can be found in Supplementary C.

### 3.3 Feature reconstruction and loss functions

The student branch aims to reconstruct the full stacked feature map, $\hat{\mathbf{F}} = D_{\text{stu}}(S) \in \mathbb{R}^{H \times W \times C_{\text{tot}}}$, while the teacher reconstructs the lower-dimensional adapted feature map, $\hat{\mathbf{F}}_{\text{adapt}} = D_{\text{tea}}(S) \in \mathbb{R}^{H \times W \times d}$. Student and teacher reconstruction losses are defined as:

$$\mathcal{L}_1 = \frac{1}{HWC_{\text{tot}}} \left\| \hat{\mathbf{F}} - \mathbf{F} \right\|_2^2 , \tag{3}$$

$$\mathcal{L}_2 = \frac{1}{HWd} \left\| \hat{\mathbf{F}}_{\text{adapt}} - \mathbf{F}_{\text{adapt}} \right\|_2^2 . \tag{4}$$

As illustrated in Fig. 1, $\mathcal{L}_1$ and $\mathcal{L}_2$ independently optimize the student and teacher branches.

**Progressive Distillation**   To prevent collapse (i.e., trivial reconstructions to zero target), we adopt a progressive training strategy. During early training, gradients from the teacher decoder to its adapter are blocked and the adapter is updated via exponential moving average (EMA) from the student. This stabilizes training, as reconstructing raw foundation features is more reliable when starting from scratch.

### 3.4 Federated Learning with Two-Stage Knowledge Transfer

**Stage 1: Federated representation distillation and alignment**   When starting from scratch (early rounds), as described in Algorithm 1, clients send updated *student adapter* and *slot encoder* parameters to the server. The server aggregates these using FedAvg:

$$\Theta^{(r+1)} = \frac{1}{\sum_c n_c} \sum_c n_c \, \Theta_c^{(r)} \tag{5}$$

where $\Theta$ includes the adapter and slot encoder weights. This enables learning a global representation adapter that generalizes across all client datasets, along with a global Slot Attention module for object discovery across domains. In this stage only the student will learn to optimize the adapter, while the teacher will learn a more adapted, domain-specific object-discovery.

**Stage 2: Cross-branch federation.**   In later rounds, the *teacher* branch is allowed to further optimize its own adapter and Slot Attention module. The refined knowledge is then transferred to the student via a self-averaging strategy we term *Local FedAvg*. For this step, conventional knowledge distillation (KD) is challenging due to Slot Attention's permutation invariance. Slot alignment via Hungarian matching [38] could make attention-based KD feasible, however we adopt a simpler

**Algorithm 1** Federated Two-Branch Slot Attention Training in FORLA

1: **Initialize:** Global parameters $\Theta^0 = (\phi^0, \theta^0)$ for adapter $g_\phi$ and Slot Attention $\text{SA}_\theta$
2: **Initialize:** Local teacher branch $(\phi_{\text{tea}}^0, \theta_{\text{tea}}^0) \leftarrow (\phi^0, \theta^0)$ via EMA
3: **for** each round $r = 1, 2, \ldots, R$ **do**
4:     **for** each client $c \in \{1, \ldots, C\}$ **in parallel do**
5:         Load global model: $(\phi_c, \theta_c) \leftarrow (\phi^{(r-1)}, \theta^{(r-1)})$         # synchronize local models
6:         **for** each local epoch $e = 1$ to $E$ **do**
7:             **for** each batch $B$ in client $c$'s data **do**
8:                 Extract and concatenate features $F$ from $M$ frozen foundation models
9:                 Compute adapted features $\mathbf{F}_{\text{adapt}} = g_{\phi_c}(F)$         # learnable adapter
10:               Obtain slots $S = \text{SA}_{\theta_c}(\mathbf{F}_{\text{adapt}})$         # learnable slot attention
11:               **Student loss:** $\mathcal{L}_1 = \|\hat{F} - F\|_2^2$ where $\hat{F} = D_{\text{stu}}(S)$ (Eq. 3)
12:               **Teacher loss:** $\mathcal{L}_2 = \|\hat{F}_{\text{adapt}} - \mathbf{F}_{\text{adapt}}\|_2^2$ where $\hat{F}_{\text{adapt}} = D_{\text{tea}}(S)$ (Eq. 4)
13:               Update student branch $(\phi_c, \theta_c)$ using $\mathcal{L}_1$         # gradient update for student
14:               **if** in later rounds **then**
15:                   Update teacher branch $(\phi_c^{\text{tea}}, \theta_c^{\text{tea}})$ using $\mathcal{L}_2$
16:                   **Local FedAvg :** $(\phi_c, \theta_c) \leftarrow \alpha(\phi_c, \theta_c) + (1-\alpha)(\phi_c^{\text{tea}}, \theta_c^{\text{tea}})$       # local FedAvg
17:               **else**
18:                   Update teacher branch $\theta_c^{\text{tea}}$ using $\mathcal{L}_2$
19:                   **EMA Update:** $\phi_c^{\text{tea}} \leftarrow \text{EMA}(\phi_c)$   # early stage: update teacher via EMA
20:               **end if**
21:             **end for**
22:         **end for**
23:         Send updated student branch $(\phi_c, \theta_c)$ to server         # upload local parameters
24:     **end for**
25:     **Global aggregation:** $(\phi^{(r)}, \theta^{(r)}) \leftarrow \text{FEDAVG}(\{\phi_c, \theta_c\}_{c=1}^C)$      # next-round global model
26: **end for**
27: **Return:** Global model $(\phi^R, \theta^R)$

approach. We treat the student and the teacher as two local clients, sharing knowledge through weight averaging using FedAvg. This allows the teacher to be indirectly globalized by synchronizing with the student's Slot Attention module.

In such a framework, the KD aspect (the two-branch reconstruction) is tightly integrated with FL: each client's training is effectively distilling the knowledge of multiple foundation models into the shared slot representation, and, in return, FL can benefit from the knowledge transfer between teacher and student. By aggregating the adapter and slot encoder, a *universal* representation adapter and an object discovery module are learned by leveraging knowledge across all domains.

## 4   Experiments and Results

We evaluate FORLA on a wide range of visual domains. Through extensive experiments, we aim to address the following research questions: 1) **Centralized vs. decentralized learning**: Can federated object-centric learning match or exceed centralized performance? 2) **Representation adapters**: Which feature adaptation and distillation strategies work best across training regimes? 3) **Algorithm compatibility**: Can FORLA integrate with various federated optimization algorithms beyond FedAvg, such as FedProx and FedAdam? 4) **Representation quality**: How interpretable are FORLA's representations compared to frozen foundation features? 5) **Impact of domain gap**: How does the domain gap across datasets affect object discovery in solo, centralized, and federated training setups? 6) **Component contribution**: What is the role of each design component—distillation, adapter, and personalization in the overall performance?

We perform rigorous evaluations on both surgical and natural datasets using object discovery metrics. FORLA outperforms individual and centralized training, yielding interpretable representations and scene segmentation capabilities without supervision or data sharing.

**Dataset**   We evaluate our experiment on seven datasets, categorized into two groups: 1) **Surgical vision datasets**: including the abdominal surgical dataset [72], Cholec80 dataset [58], and a proprietary thoracic surgery dataset. 2) **Natural vision datasets**: including COCO [36], PASCAL VOC 2012 [13], YouTube-VIS [66], and YouTube-Objects [47]. In total, these datasets comprise

Table 1: Federated performance on surgical and natural data (stratified evaluation).

| Adapter | Method | Surgical | | | | | | | | | Natural | | | | | | | | | |
|---|---|---|---|---|---|---|---|---|---|---|---|---|---|---|---|---|---|---|---|---|---|
| | | Abdominal | | | Cholec | | | Thoracic | | | COCO | | | PASCAL | | | YTVIS | | | YTOBJ | |
| | | mBO | FG-ARI | CorLoc | mBO | FG-ARI | CorLoc | mBO | FG-ARI | CorLoc | mBO | FG-ARI | CorLoc | mBO | FG-ARI | CorLoc | mBO | FG-ARI | CorLoc | mBO | CorLoc |
| DINO | Individual | 47.33 | 57.87 | 52.38 | 28.7 | 38.9 | 30.4 | 30.96 | 19.3 | 30.45 | 23.96 | 29.2 | 20.13 | 34.79 | 35.17 | 52.31 | 33.09 | 36.64 | 54.02 | 42.77 | 54.78 |
| SAM | Individual | 47.00 | 53.70 | 56.20 | 25.75 | 32.62 | 31.81 | 50.28 | 41.14 | 54.88 | 22.61 | 25.02 | 20.50 | 36.98 | 35.78 | 35.78 | 32.62 | 36.07 | 36.07 | 42.86 | 54.95 |
| Concat | Individual | 48.65 | 54.07 | 64.30 | 28.68 | 37.96 | 33.89 | 48.77 | 40.56 | 51.19 | 22.49 | 25.06 | 47.98 | 35.58 | 35.24 | 47.25 | 32.97 | 35.97 | 53.28 | 43.33 | 50.56 |
| MLP | Individual | 51.85 | 59.04 | 69.25 | 33.54 | 43.59 | 53.8 | 54.0 | 43.59 | 58.0 | 24.41 | 27.63 | 57.67 | 35.77 | 35.0 | 51.65 | 35.16 | 38.25 | 58.08 | 48.62 | 60.31 |
| | Centralized | 55.91 | 61.31 | 76.47 | 31.12 | 41.12 | 43.37 | 54.44 | 44.2 | 63.06 | 25.12 | 27.99 | 60.1 | 36.59 | 35.81 | 52.2 | 34.47 | 37.5 | 54.44 | 46.67 | 57.43 |
| | FORLA | 56.71 | 62.81 | 79.73 | 34.58 | 44.75 | 50.85 | 59.8 | 47.02 | 70.45 | 25.62 | 26.84 | 60.58 | 38.32 | 36.46 | 66.48 | 36.51 | 40.14 | 57.35 | 46.49 | 57.52 |
| MOE | Individual | 51.83 | 57.54 | 72.02 | **34.64** | 45.12 | 52.42 | 51.1 | 41.87 | 55.89 | 24.69 | 28.32 | 60.9 | 34.36 | 32.99 | 52.75 | 34.82 | 38.38 | 56.33 | 48.51 | 61.52 |
| | Centralized | 54.89 | 60.66 | 75.58 | 30.37 | 40.28 | 42.69 | 55.6 | 44.8 | 64.57 | 24.82 | 27.72 | 57.51 | 36.72 | 36.21 | 51.1 | 34.69 | 38.08 | 52.4 | 44.86 | 53.43 |
| | FORLA | 56.42 | 61.95 | 75.45 | 34.06 | 45.26 | 49.14 | 57.71 | 46.12 | 63.17 | 26.65 | 28.7 | 65.59 | 39.43 | 37.44 | 64.84 | 39.38 | 43.42 | 63.9 | 48.4 | 62.21 |
| AFM | Individual | 50.34 | 59.04 | 70.73 | 32.9 | 42.46 | 42.36 | 53.03 | 43.48 | 56.62 | 24.15 | 27.76 | 56.22 | 34.98 | 34.16 | 51.1 | 33.94 | 37.54 | 54.59 | 48.98 | 65.11 |
| | Centralized | 55.26 | 61.5 | 77.97 | 32.87 | 42.92 | 47.36 | 55.93 | 44.62 | 65.58 | 23.73 | 26.97 | 56.06 | 37.16 | 36.4 | 54.4 | 33.47 | 36.9 | 51.24 | 44.51 | 52.13 |
| | FORLA | **57.86** | **64.88** | **80.3** | 34.2 | **44.35** | **54.49** | **61.86** | **47.58** | **75.42** | **27.22** | **30.11** | **64.94** | **40.83** | **39.51** | 64.84 | **38.51** | **43.08** | **64.92** | **51.92** | **66.87** |

Table 2: Federated performance on mixed-domain training with all datasets.

| Method | Abdominal | | Cholec | | Thoracic | | COCO | | PASCAL | | YTVIS | | YTOBJ | | Average | |
|---|---|---|---|---|---|---|---|---|---|---|---|---|---|---|---|---|
| | mBO | CorLoc | mBO | CorLoc | mBO | CorLoc | mBO | CorLoc | mBO | CorLoc | mBO | CorLoc | mBO | CorLoc | mBO | CorLoc |
| Individual + AFM | 50.34 | 70.73 | 32.90 | 42.36 | 53.03 | 56.62 | 24.15 | 56.22 | 34.98 | 51.10 | 33.94 | 54.59 | 48.98 | **65.11** | 40.33 | 56.82 |
| Centralized + MLP | 49.78 | 58.08 | 25.49 | 34.11 | 44.83 | 41.95 | 24.54 | 56.54 | 36.71 | 53.85 | 33.71 | 52.11 | 44.42 | 47.74 | 37.21 | 49.34 |
| Centralized + MOE | 53.46 | 68.92 | 25.84 | 32.58 | 48.06 | 46.85 | 25.28 | 58.64 | 37.38 | 50.55 | 36.06 | 54.59 | 46.05 | 53.34 | 39.73 | 52.21 |
| Centralized + AFM | 54.16 | 72.5 | 31.64 | 46.42 | 53.72 | 57.49 | 26.68 | 62.20 | 38.93 | 58.93 | 37.15 | 60.41 | 47.84 | 57.41 | 41.45 | 58.81 |
| Centralized + Individual decoder | 54.87 | 70.88 | 27.39 | 37.83 | 50.91 | 52.39 | 25.98 | 62.20 | 39.09 | 59.34 | 36.30 | 57.21 | 47.41 | 58.62 | 40.28 | 56.78 |
| Slot FedAvg + MOE | 55.77 | 70.55 | 30.23 | 44.27 | 54.53 | 59.93 | 26.37 | 63.97 | **39.81** | 63.19 | 37.78 | 61.86 | 48.98 | 63.65 | 41.92 | 61.20 |
| Slot FedAvg + AFM | 53.74 | 69.65 | 32.63 | 41.65 | 52.01 | 50.42 | 25.45 | 63.17 | 39.13 | 63.74 | 37.60 | 62.30 | 48.12 | 61.74 | 41.24 | 58.95 |
| FORLA + MLP | 56.54 | **77.68** | 33.36 | **50.71** | **58.36** | **68.32** | 24.75 | 56.22 | 37.54 | 51.65 | 36.91 | 57.50 | 44.25 | 52.18 | 41.53 | 59.04 |
| FORLA + MOE | 56.03 | 73.08 | 31.78 | 42.43 | 56.79 | 62.73 | 26.41 | 62.84 | 38.29 | 60.44 | 37.43 | 57.35 | 49.33 | 63.79 | 42.29 | 60.37 |
| FORLA + MOE + FedProx | 54.19 | 72.62 | 32.42 | 44.06 | 57.24 | 61.81 | **26.73** | 62.52 | 39.19 | 60.44 | 37.87 | 60.84 | 46.27 | 57.95 | 42.13 | 59.89 |
| FORLA + MOE + FedAdam | 55.02 | 73.50 | 32.58 | 43.14 | 55.63 | 59.82 | 26.28 | 63.00 | 39.05 | 57.69 | 37.27 | 57.93 | 47.34 | 58.99 | 41.88 | 59.01 |
| FORLA + AFM | **61.84** | 77.32 | **33.96** | 44.40 | 56.74 | 59.17 | 26.45 | **64.46** | 38.02 | **65.38** | **39.33** | **67.39** | 49.36 | 64.02 | **43.81** | **63.02** |
| FORLA Surgery \| Natural | 57.86 | 80.30 | 34.20 | 54.49 | 61.86 | 75.42 | 27.22 | 64.94 | 40.83 | 64.84 | 38.51 | 64.92 | 51.92 | 66.87 | 44.63 | 67.40 |

approximately 1.4 million images. With the exception of the proprietary thoracic dataset, all data is publicly available.[1] Further details are provided in Supplementary Material E.

**Experiment setup** Following other research on object centric representation learning papers (e.g MONET [5], IODINE [18] and FT-DINO [11]), we evaluate our approach based on the quality of the slot attention masks using four primary metrics: Foreground Adjusted Rand Index (FG-ARI) [17], Mean Best Overlap (mBO) [46], and Correct Localization (CorLoc) [3] which are widely adopted in object-centric research [52, 39, 11]. We also compute Mean Best Hausdorff Distance (mBHD), which captures boundary-level accuracy and is particularly important for clinical data.

We compare three training regimes: (i) *Individual*, where datasets are trained separately; (ii) *Centralized mixed*, where related domains are combined; and (iii) *Federated*, where each dataset acts as an isolated client (up to seven domains). Features are extracted from frozen ViT-B/16 encoders of DINO, MAE, CLIP (vision branch only), and SAM (encoder for feature only). Each client trains for 100+ epochs with early stopping after 30 stagnant epochs. FedAvg is performed globally every 100 iterations and locally (student-teacher) every 1000. We use Adam (lr = $4\times10^{-4}$, batch size = 16). Please see more implementation details in the Supplementary Material F. Unless noted, evaluation uses the student decoder, while teacher results appear in Supplementary G.

**Main results** Given a learned global representation via a feature adapter and a global object-centric slot attention module, we first investigate how their performance compares to models trained on individual datasets and centralized models trained on mixed datasets assigned to each client. We begin with a controlled setting where domain gaps are smaller: using three clients for surgical data and four clients for natural datasets in a federated setup. These results are summarized in Table 1.

We explore three types of adapters: MLP, MOE, and AFM, under individualized, centralized, and federated training. We also experiment using single foundation models and a naive feature stack (denoted as "Concat" in Table 1) without adaptation. Among the single foundation models, DINO

---

[1]An open-access version of the proprietary thoracic dataset is curated as part of the federated learning benchmark in this work.

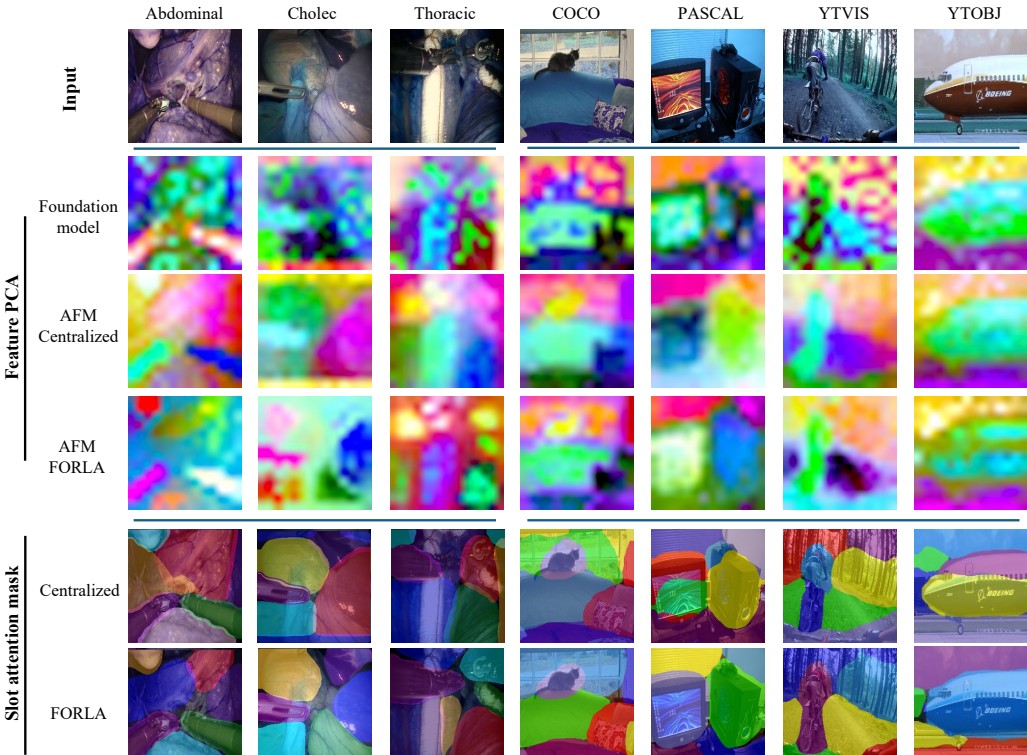

Figure 2: Visualization of PCA maps and Slot Attention (SA) masks from different methods. The middle three rows show the first three PCA components visualized using RGB channels for frozen foundation model features and adapted features produced by the AFM module trained under centralized training and under FORLA. The last two rows illustrate the scene decomposition ability of each method via the SA-generated masks.

and SAM achieved the best overall performance. Results for CLIP and MAE are provided in the Supplementary Material G. Interestingly, concatenated features slightly outperform others on surgical datasets (e.g. achieving 64.3 CorLoc on Abdominal), and are comparable to SAM and DINO on natural datasets, likely due to their shared pretraining on natural images, enabling complementarity when transferred to new domains. Once adapters are introduced, slot attention shows accuracy gains across all domains. Regardless of adapter type or training approach, performance improves. Among training strategies, the proposed federated method consistently outperforms both individualized and centralized training. Within surgical datasets, performance gains on Abdominal and Thoracic are especially prominent. FORLA with AFM reaches 80.3 CorLoc on Abdominal and 75.42 on Thoracic, whereas Cholec achieves smaller gains due to closer domain proximity and distinct instrumentation.

For natural datasets, federated training also yields consistent performance improvements. Interestingly, for some subdomains, centralized training underperforms individualized training, likely because unsupervised slot attention benefits from domain-specific curation [11]. For instance, YTOBJ attains 48.98 mBO with individual training but drops to 44.51 mBO under centralized training using the same AFM adapter. Since YTOBJ has only 10 object categories but abundant data, mixing it with other datasets may dilute the inherent saliency, making unsupervised learning harder. This may also explain why most state-of-the-art slot attention work avoids mixed training. Among adapters, the attention-based AFM performs best, likely due to its scalability and ability to preserve rich foundation feature channels, achieving top CorLoc scores of 66.87 on YTOBJ and 64.94 on PASCAL (Table 1).

To better understand how different approaches learn representations, Figure 2 shows the top three PCA projections of features visualized using RGB channels. We observe that FORLA captures more fine-grained and globally coherent semantic structures compared to centralized mixed training. For instance, it more effectively separates distinct tissue textures in surgical scenes and groups semantically related object parts holistically. Leveraging these adapted representations, the globally

Table 3: Study on the Self-Distillation on Different Datasets and Setups.

| Method | Abdominal | | | | Cholec | | | | Thoracic | | | |
|---|---|---|---|---|---|---|---|---|---|---|---|---|
| | mBO | mHD ↓ | FG-ARI | CorLoc | mBO | mHD ↓ | FG-ARI | CorLoc | mBO | mHD ↓ | FG-ARI | CorLoc |
| Individual + distill 1 stage | 33.91 | 80.240 | 35.41 | 29.22 | **35.35** | **56.349** | **45.28** | **48.89** | 33.25 | 109.666 | 22.70 | 30.07 |
| Individual + distill 2 stages | 56.93 | 36.231 | 63.52 | **79.40** | 32.50 | 62.133 | 41.98 | 46.47 | 49.44 | 77.084 | 41.54 | 55.14 |
| Centralized + distill | 54.34 | 50.635 | 59.34 | 68.43 | 27.82 | 76.464 | 36.41 | 37.14 | 48.73 | 77.987 | 40.61 | 47.07 |
| FORLA self distill | **61.84** | **33.909** | **69.49** | 77.32 | 33.96 | 59.801 | 45.07 | 44.40 | **56.74** | **59.790** | **45.42** | **59.17** |

| Method | COCO | | | PASCAL | | | YTVIS | | | YTOBJ | | |
|---|---|---|---|---|---|---|---|---|---|---|---|---|
| | MBO | FG-ARI | CorLoc | MBO | FG-ARI | CorLoc | MBO | FG-ARI | CorLoc | MBO | FG-ARI | CorLoc |
| Individual + distill 1 stage | 21.78 | 24.5 | 47.66 | 33.77 | 34.31 | 53.85 | 33.97 | 36.61 | 53.57 | **50.8** | **46.83** | **65.39** |
| Individual + distill 2 stages | 24.56 | 27.5 | 53.8 | 36.09 | 35.26 | 50.00 | 35.26 | 36.03 | 46.87 | 50.37 | 46.61 | 63.97 |
| Centralized + distill | 25.77 | 28.99 | 60.9 | 37.90 | 37.34 | 56.04 | 35.15 | 38.6 | 54.59 | 46.43 | 42.67 | 55.46 |
| FORLA self distill | **26.45** | **30.21** | **64.46** | **38.02** | **39.44** | **65.38** | **39.33** | **45.08** | **67.39** | 49.36 | 45.77 | 64.02 |

shared SA module is able to explicitly decompose scenes into meaningful object-level masks, as illustrated in the last two rows of Figure 2.

We further test a more challenging setting by mixing all 7 datasets (surgical + natural) across 7 federated clients, using the AFM FORLA model trained under lower-heterogeneity (stratified) settings as a strong baseline to see whether federated or fully mixed centralized training can surpass it. As shown in Table 2, only the centralized model with AFM surpasses the individualized baseline; for instance, Centralized+MLP achieves 49.34 CorLoc versus 56.82 for Individual+AFM, likely due to limited decoder scalability. To test whether limited decoder scalability caused centralized underperformance in comparison to FL, we experiment a hybrid setup: initializing a centralized model with AFM, duplicating it to each client, freezing the adapter and slot attention, and fine-tuning individual decoders per client. While this improves performance on some datasets like Abdominal and PASCAL, it reduces performance on others, lowering the overall average, rejecting our hypothesis.

All federated approaches tend to outperform centralized or individualized training, thanks to the inherent domain disentanglement in object-centric learning. Even if we remove the teacher from FORLA and apply standard FedAvg to the student's adapter and slot attention (referred to as *slot FedAvg*) we see improvements over centralized and individualized baselines. Our full model (FORLA) with the AFM adapter achieves the highest performance despite the substantial heterogeneity across sub-domains. Interestingly, slot FedAvg performs well on the Natural dataset when using a MoE adapter but struggles on Surgical datasets. This discrepancy likely arises because foundation models are typically pretrained on natural data, making teacher supervision and feature adaptation more critical for domains like Surgical, which are underrepresented in pretraining. It is noteworthy that MLP adapts well to surgical domains (e.g., 79.73 CorLoc on Abdominal in Table 1), but performance drops on natural domains (e.g., 52.18 on YTOBJ in Table 2). MOE lies in between, consistent with its moderate gains. We also evaluated FedProx and FedAdam within FORLA+MOE that help in specific domains but do not yield general performance boosts (Table 2).

In conclusion, FORLA remains most effective when applied separately to groups of clients holding sub-domain data from either surgical or natural domains. Nevertheless, even under increased heterogeneity, FORLA remains effective and outperforms both individualized and centralized training, highlighting its strength in learning disentangled, generalizable representations without data sharing.

**Effect of Self-Distillation on Different Datasets and Setups**  As demonstrated, FORLA outperforms direct FedAvg on adapter and slot attention, mainly due to its teacher–student design for feature distillation and two-stage reconstruction. To test whether this also benefits individualized or centralized training, we applied the same strategy to both. For individualized training, two versions are compared: a one-stage setup (EMA from scratch) and a two-stage setup (EMA followed by distillation). In most datasets, the two-stage version outperforms the one-stage setup. For example, as shown in Table 3, on Abdominal, mBO improves from 33.91 (1-stage) to 56.93 (2-stage), and CorLoc from 29.22 to 79.40. However, for Cholec and YTOBJ, the one-stage setup performs strongly (e.g., YTOBJ: 65.39 CorLoc), possibly because slot attention in these datasets relies less on foundation features—a pattern also seen in simple-scene slot attention literature [38].

For centralized training, the two-stage approach is on par with no-distillation setups (Table 2) gaining moderate accuracy on natural data, and far below FORLA's federated version. This observation

Table 4: Further model personalization experiment.

| | Abdominal | | | Cholec | | | Surgical Avg | | | YTVIS | | YTOBJ | | Natural Avg | |
|---|---|---|---|---|---|---|---|---|---|---|---|---|---|---|---|
| Method | mBO | FG-ARI | CorLoc | mBO | FG-ARI | CorLoc | mBO | FG-ARI | CorLoc | mBO | CorLoc | mBO | CorLoc | mBO | CorLoc |
| Individual | 51.85 | 59.04 | 69.25 | 33.54 | 43.59 | 53.8 | 42.7 | 51.32 | 61.53 | 35.16 | 58.08 | 48.62 | 60.31 | 41.89 | 59.2 |
| Centralized | 54.16 | 59.54 | 72.5 | 31.64 | 41.5 | 46.42 | 42.9 | 50.52 | 59.46 | 37.15 | 60.41 | 47.84 | 57.41 | 42.5 | 58.91 |
| FORLA — No personalization | 57.04 | 63.82 | 74.75 | 32.38 | 42.54 | 48.44 | 44.71 | 53.18 | 61.6 | 37.72 | 59.53 | 46.93 | 59.71 | 42.33 | 59.62 |
| FORLA — Personalized SA | **60.21** | 66.75 | **78.25** | 34.04 | 45.05 | 51.29 | 47.13 | 55.9 | 64.77 | 38.4 | 64.34 | **48.57** | **62.23** | 43.49 | 63.29 |
| FORLA — Personalized adapter | 59.63 | **66.8** | 77.63 | **34.86** | **45.14** | **54.14** | **47.25** | **55.97** | **65.89** | **38.79** | **65.21** | 48.45 | 61.74 | **43.62** | **63.48** |

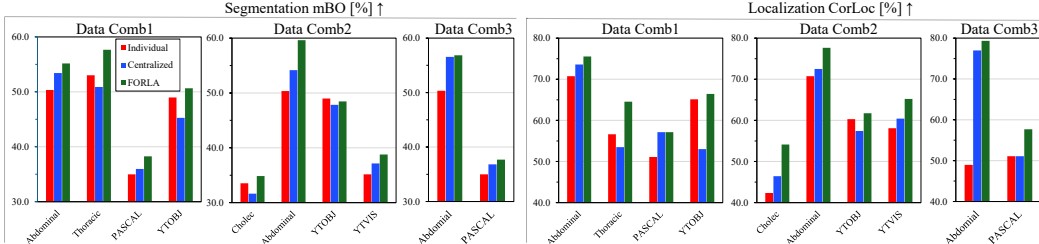

Figure 3: mBO and Corloc for centralized training and FORLA across different data combinations. Data Comb: Data combinations. Individualized training is presented as baseline here.

Table 5: Results of combining two surgical domains and distributing across clients.

| Method | Abdominal | | | Thoracic | | |
|---|---|---|---|---|---|---|
| | mBO | FG-ARI | CorLoc | mBO | FG-ARI | CorLoc |
| Single domain [50%×2] | 50.34 | 59.04 | 70.73 | 53.03 | 43.48 | 56.62 |
| Centralized [2×50%×2] | 56.56 | 62.80 | 76.97 | 54.52 | 44.37 | 62.67 |
| FORLA | **57.30** | **63.30** | **77.08** | **61.55** | **47.84** | **71.53** |

Table 6: Results of different data partitions within the same domain (one domain with 25% or 10% ×4).

| Method | PASCAL [25%] | | Thoracic [25%] | | PASCAL [10%] | | Thoracic [10%] | |
|---|---|---|---|---|---|---|---|---|
| | mBO | FG-ARI | mBO | FG-ARI | mBO | FG-ARI | mBO | FG-ARI |
| Individual [25%/10%] | 33.21 | 31.52 | 46.39 | 39.79 | 30.71 | 28.11 | 27.14 | 23.35 |
| Centralized [25/10% ×4] | 34.98 | 34.16 | 53.03 | 43.48 | **34.99** | **33.52** | 49.17 | 40.14 |
| Slot-FedAvg | 35.16 | 32.94 | **56.48** | **45.26** | 31.50 | 27.52 | **51.21** | **40.91** |
| FORLA | **35.25** | **34.14** | 54.26 | 44.30 | 32.09 | 29.25 | 49.66 | 39.21 |

further validates the advantage of the proposed framework, which tightly integrates distillation and federated training by treating the teacher and student as independent local clients.

**Further Personalization Based on Global Representation** We also explored whether further personalization further improves object discovery after learning global features representation and slot attention. Under a stratified setup (excluding Thoracic, COCO, and PASCAL), as shown in Table 4, FORLA (without personalization) already outperforms individualized and centralized models in this configuration. Next, we initialize models with the global federated version and personalize them by freezing either the adapter or slot attention, allowing only one of them to be optimized locally. Both setups show that domain heterogeneity can be further countered by personalization while still leveraging shared knowledge. This is particularly effective for surgical data, which foundation models have been rarely trained on.

**Robustness to Data Partitioning and Domain Combinations** We further evaluate the robustness of FORLA under various data stratification and partitioning schemes. Figure 3 shows that FORLA consistently outperforms individualized and centralized models across two- and four-domain combinations. In cases with large domain gaps (e.g., surgical vs. natural), centralized training can even underperform individualized setups (e.g., Thoracic in Comb1, PASCAL in Comb2), while FORLA scales effectively across domains. Table 5 presents results where the Abdominal and Thoracic datasets are each split between two clients (50% per client). While the benefit of exploiting structural coherence within a local dataset diminishes when the data is distributed, our federated approach still surpasses centralized training, particularly on the Thoracic domain.

We also test performance under data scarcity by partitioning a single domain across four clients (Table 6). We first split a full dataset (25% per client) and then a reduced dataset (10% per client, totaling 40% of the original). The results indicate that the advantage of federated object-centric learning declines as the data per client becomes more limited. This trend is consistent across both FORLA and the simpler Slot-FedAvg. This highlights the importance of having sufficiently large and representative local datasets for unsupervised object-centric learning to discover meaningful concepts.

Table 7: Comparison to zero-shot segmentation of full SAM models with mask decoders.

| | PASCAL | | | Abdominal | | | Thoracic [†] | | |
|---|---|---|---|---|---|---|---|---|---|
| | mBO | FG-ARI | CorLoc | mBO | FG-ARI | CorLoc | mBO | FG-ARI | CorLoc |
| SAM (ViT-B) | 23.79 | 24.55 | 35.16 | 45.76 | 52.10 | 39.15 | 18.10 | 16.41 | 8.35 |
| SAM (ViT-H) | **54.90** | **57.31** | **79.12** | **66.66** | **69.81** | 65.47 | 33.57 | 30.48 | 23.73 |
| FORLA | 40.83 | 39.51 | 45.04 | 57.86 | 64.88 | **80.30** | **61.86** | **47.58** | **75.42** |

[†] Note that Thoracic is less publicly available thus likely not included in the training of SAM.

These findings are consistent with the general behavior of object-centric models, confirming that our framework requires data not to be overly partitioned.

**Comparison to Supervised Segmentation Foundation Model**  We compare the unsupervised segmentation of FORLA to the zero-shot segmentation of full SAM models (using ViT-B and the ViT-H with its mask decoder trained under supervision). We found that SAM ViT-H is able to surpass slot attention models trained with FORLA on the natural images but fails on the very specific surgical domain (Thoracic). The underlying reason could be that the thoracic data is not publicly available and not included in the training data of SAM. In addition, the inference time for SAM ViT-H is 1.1s per image (0.85s for ViT-B), while the slot attention model takes only 3.75ms. Another advantage of OCL over SAM is that training SAM on new domains would require mask supervision and, most importantly, SAM will not capture the objectness (over decompose object parts to tiny masks) unless instructed to do so via annotation.

## 5  Conclusion

In this paper we proposed FORLA, addressing the critical challenge of learning disentangled visual representations from distributed, non-IID datasets. By integrating object-centric inductive biases with federated optimization FORLA enables collaborative learning across heterogeneous domains preserving local data structure. Extensive experiments across seven surgical and natural vision datasets demonstrate FORLA's superiority over both individualized and centralized learning of object representation. By using light adapters to learn from cached foundation features distributively, FORLA is an economical framework for large scale object-centric representation learning without centralized data or heavy compute infrastructure, achieving a 6.7× reduction in communication cost.

**Limitations.**  While FORLA shows strong potential in learning universal object-centric representations from heterogeneous data, several limitations remain. First, real-world visual scenes often exhibit hierarchical compositionality, where objects are nested or composed of multiple parts at varying levels of granularity. Our current framework still assumes that there is an optimal decomposition hierarchy according to the given dataset, but such compositionality should be downstream task-specific. Second, our approach is still built on foundation models and struggles to discover objects purely based on image reconstruction [52], as it relies on image properties that have already been captured by foundation models.

**Future work.**  Several research directions can further enhance FORLA. Incorporating a dynamic slot mechanism [14, 34] will automatically adjust the number of slots per scene. We also intend to explore alternative decoders, such as slot-diffusion decoder [25], to improve reconstruction and flexibility, as well as high resolution encoders such as DINOv3 [55]. Extending FORLA to support both image and video data [33] would enable unified training from mixed temporal modalities. Our approach could also support a federated object-centric foundation model for downstream tasks such as video action recognition, using slot representations as region-based tokens [54]. Integrating weak supervision signals from classification tasks [35] or language prompts [65] available to only a subset of clients could enhance the semantic alignment of learned slots and bridge the gap between vision and language in federated settings. In addition, incorporating additional learning objectives tailored to dataset subtypes or tasks, while using slots to bridge representations, could lead to more universal, task-agnostic representations [32]. Another promising next step could involve experimentation on robotic manipulation data [27]. These datasets represent a mix of data with specialized actions, similar to robotic surgery, and Natural datasets, as they act on everyday objects.

## Acknowledgments and Disclosure of Funding

This work was supported by the Linda Pechenik Montague Investigator Award and the American Surgical Association Foundation Fellowship.

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

# Supplementary Materials for FORLA: Federated Object-Centric Representation Learning with Slot Attention

## A  Extended Related work

To provide additional context for our approach, we expand on key areas of related work in this section.

### A.1  Federated Learning (FL)

FL enables collaborative training across decentralized datasets while preserving privacy [42]. Prior work addresses challenges like non-IID data [31] and communication efficiency [29], but most focus on classification or segmentation tasks without emphasis on learning representations. Early work on unsupervised representation learning for FL showed that naive FedAvg misaligns local feature spaces under non-IID data. Zhang et al. [69] addressed this with FURL/FedCA, which maintains a shared memory bank and a contrastive alignment module. Michieli et al. [44] proposed Prototype-Guided FL, where clients periodically exchange class-agnostic prototypes to pull their feature spaces into a common geometry. Tan et al. [57] also leverage prototypes in FedPCL in a contrastive learning scheme. This idea is later extended by clustering-based methods such as ORCHESTRA [40] and FedRLC [43]. Knowledge distillation has also emerged as a powerful alignment tool. FedX [19] employs two-sided knowledge distillation with a contrastive objective, allowing clients to share rich representation information without exchanging data while FedUKD [63] further reduces domain bias via bilateral distillation in an unsupervised FL framework. Despite this progress, object-centric representation learning – which aims to decompose scenes into entity-specific features – has not yet been explored in federated settings, and no prior work explicitly combines these paradigms to date, despite its potential to leverage domain-specific data.

### A.2  Object-Centric Learning

Slot Attention (SA) [38] pioneered the use of iterative attention to decompose scenes into object slots. Subsequent work improved its scalability [56] and incorporated transformers [51], but all assume centralized training on mixed data. ORL [64] demonstrated that exploiting object-level correspondences can improve unsupervised learning on multi-object images. Following the introduction of DINOSAUR [52], SA began to be applied to real-world scenarios using foundation models. A growing body of work has since built on this approach to improve performance—either by augmenting foundation features [39], or adding modality [67, 2], or by fine-tuning the foundation models themselves [12]. Recent techniques like contrastive learning objective [41], patch permutation augmentation [26] and using language supervision signals [10] have been demonstrated to improve the scene decomposition performance. However, no prior work in federated learning has combined object-centric models with distributed training across heterogeneous datasets. We argue that centralized training of object-centric models struggles with concept entanglement, a challenge that federated learning inherently mitigates by preserving local data coherence.

### A.3  Concept Entanglement in Multi-Domain Learning

Concept entanglement represents a significant challenge in the field of multi-domain learning, particularly when training machine learning models on datasets that contain overlapping visual semantics. Early work tackles this problem by explicitly splitting features into shared and private subspaces, as in Domain Separation Networks [4]. Subsequent approaches refine the idea: [37] propose a Unified Feature Disentangler that learns domain-invariant codes via adversarial training and [24] maximizes interaction information to carve representations into domain-specific and domain-agnostic parts. Latent-domain methods reduce entanglement without domain labels—for example using a sparse adapter for Latent Domain Learning [9]. [32] show that distilling representations from multiple task- and domain-specific networks using alignment with small domain-specific adapters can lead to efficient universal representations. Despite these advances, object-centric disentanglement has yet to be explored in any centralized or federated setting, leaving a gap that our work addresses.

### A.4 Adaptation Layers and Feature Harmonization

Adaptation layers are widely used in transfer learning [49] and multi-task learning [59] to align feature spaces. In federated learning (FL), [8] employs adapters for personalized federated learning. Didolkar et al. [12] investigate the transferability of foundation models in object-centric representation learning and introduce a fine-tuning framework (FT-DINOSAUR) for the task of object discovery, achieving strong unsupervised transfer. Crucially, their results suggest that object-centric learning can be improved by training on one domain and fine-tuning on another, but that improvements cannot be achieved merely by scaling up data. Our work goes further by demonstrating how federated learning can scale to heterogeneous datasets through cross-client feature harmonization, ensuring global model stability despite diverse local inputs. One technique for adapting foundation models is the application of lightweight feature adapters [16, 70], which introduce trainable components into the network while keeping most of the model frozen. Other methods include fine-tuning only the bias parameters of pretrained models [6, 68], and learning low-rank adaptations [22]. We also experiment with adaptation modules correlated to a masking approach based on frozen foundation models [60]. An advantage of this approach is that multiple foundation models can be used in parallel, without any query on which one to fine-tune.

### A.5 Knowledge Distillation

Knowledge distillation has been used to perform federated learning, for instance, when a global student calculated with FedAvg learns for a local teacher [71, 61]. In large-scale self-supervised learning, such as DINO [7], self-distillation enables the model to implicitly ensemble knowledge over time, with the teacher network being updated through an exponential moving average (EMA) of the student, akin to Polyak-Ruppert averaging [45]. Inspired by this approach, our framework updates certain modules of the teacher by averaging them with the student, but without relying on a conventional knowledge distillation (KD) loss. This is because slot prediction is inherently permutation-invariant [38], making direct teacher-to-student supervision inapplicable. Instead, we synchronize the teacher and student Slot Attention modules through periodic averaging—effectively a local FedAvg operation between the two branches. This strategy is also related to co-distillation [1], where student-teacher models distill knowledge from one another to efficiently ensemble knowledge from large-scale, distributed data. Unlike co-distillation, where student and teacher models are typically identical, we share only certain modules between the two. Moreover, our framework operates in a fully unsupervised setting.

## B Feature adapters

We experimented with three types of adapter, *MLP adapter* [50], *mixture-of-experts (MoE)* [53, 15], and *Attention-based Feature Modulation (AFM)* [23]. Extended details on those three adapters are listed below:

- *MLP adpater* [50]: a simple MLP that maps $\mathbf{F}$ to $d$ channels. This treats the concatenated features as one vector per location and learns a single set of weights to combine and reduce them.

- *Mixture-of-experts* [53, 15]: a set of $E$ expert projection layers $\{g^{(e)}\}_{e=1}^{E}$ (each a learned linear projection) corresponding to each foundation. An auxiliary gating network produces spatially varying weights $\alpha^{(e)}(u, v)$ for each foundation model at each location $(u, v)$. The adapted feature is then $\mathbf{F}_{\text{adapt}}(u, v) = \sum_{e=1}^{E} \alpha^{(e)}(u, v) \, g^{(e)}(\mathbf{F}(u, v))$. This allows the model to dynamically select different combinations of foundation features in different images.

- *Attention-based Feature Modulation* [23]: instead of combining all features, the adapter learns to *suppress or amplify* channels from $\mathbf{F}$. We implement this as a set of learnable mask parameters $\{m_c\}$ for $c = 1, \ldots, C_{\text{tot}}$ applied to $\mathbf{F}$. The mask effectively select which foundation model features to pass through for each location. Following the channel masking, a linear projection is applied to obtain $d$ channel feature $\mathbf{F}_{\text{adapt}}$.

# C   Slot Attention Models

Slot Attention [38] is an architecture for learning object-centric representations. It maps a set of $N$ input feature vectors (e.g., image pixels or CNN features) to $K$ slot vectors, each aiming to represent an object in the scene. The $K$ slots are initialized (e.g., randomly) and iteratively updated by an attention mechanism that binds slots to parts of the input. At each iteration $t = 1 \ldots T$, attention weights $W_{ik}$ are computed between each input feature $x_i$ and each slot $s_k$ (for $i = 1, \ldots, N$ and $k = 1, \ldots, K$). For example, using dot-product attention with learned projections $W^Q, W^K$ for queries/keys, one can write:

$$W_{ik} \;=\; \frac{\exp\left( \frac{(x_i W^Q)\,(s_k W^K)^\top}{\sqrt{d}} \right)}{\sum_{k'=1}^{K} \exp\left( \frac{(x_i W^Q)\,(s_{k'} W^K)^\top}{\sqrt{d}} \right)} \;, \tag{6}$$

which is a normalized attention weight indicating how much slot $k$ attends to input $i$. Using these weights, each slot is updated by aggregating the inputs:

$$s_k \leftarrow \mathrm{GRU}\Big( s_k, \; \sum_{i=1}^{N} W_{ik}\,(x_i W^V) \Big), \tag{7}$$

where $W^V$ is a learned value projection and GRU is a gating recurrent unit that combines the current slot value with the weighted input. This process is repeated $T$ times, and includes an MLP-based refinement per iteration [38]. The result is $K$ output slot vectors $\{s_k\}_{k=1}^{K}$ that are exchangeable (permutation-invariant) and ideally each slot specializes to one object or background component.

In unsupervised object discovery, Slot Attention is typically trained by reconstructing the original image from the slots. A decoder (e.g., a CNN, spatial broadcast decoder, or transformer decoder) uses the slots to output $K$ component reconstructions and masks, which are combined to match the input image. The model thus learns to partition the scene into objects. However, such purely unsupervised training can struggle on complex real-world images. Thus, recent work of applying slot attention to such images is all based on foundational models [52, 39, 67, 11], and the reconstruction target is feature map instead of image.

# D   Additional Details on FORLA

## D.1   Two-stage Training and Local-FedAvg

We extend the FORLA framework details in Algorithm 1. The FedAvg between clients is performed every 100 iterations, while the FedAvg between teacher and student is performed every 1000 iterations. We use Adam optimizer with learning rate of $4 \times 10^{-4}$, and weight decay of $4 \times 10^{-4}$. For all data the batch size is 16.

We set an empirical switching criteria as 90K iterations, as this number allows the reconstruction loss of slot attention models on clients's student branch to converge to plateau. We also found that training for a larger number of iterations before switching from EMA (larger than 90K) can lead to better initial convergence of the student branch, but in the long term this advantage is offset by the second stage training of the student, making the additional time spent on initial training obsolete. In future, the empirical threshold can be replaced by a dynamic threshold based on tracking of student loss.

## D.2   Foundation Model Specifications

In computer vision community, a significant amount of time and resources have been used to train foundation models from large image datasets of everyday scenes and objects (*Natural vision domain*). We provide specifications of four vision foundation models used in FORLA. These four models were selected because of their complementary capabilities in semantic understanding, segmentation, reconstruction, and multimodal alignment. Table 14 summarizes key technical specifications, while we elaborate on their architectural and functional characteristics below:

- **DINO (self-DIstillation with NO labels) [7]:**

- *Architecture*: Vision Transformer (ViT-B/16, ViT-B/14 for DINO-v2 ) with 12 layers, 768D embeddings
- *Pretraining Objective*: Self-supervised distillation using image augmentations without labels
- *Training Data*: ImageNet-1k (1.28M images)
- *Key Strengths*: Captures high-level semantic relationships through global self-attention; produces spatially consistent feature maps ideal for object discovery

- **SAM (Segment Anything Model) [28]:**
  - *Architecture*: ViT-B/16 image encoder with mask decoder
  - *Training Objective*: Supervised promptable segmentation on 1B+ masks
  - *Training Data*: SA-1B dataset (11M licensed images)
  - *Key Strengths*: Specialized in boundary-aware feature extraction.

- **MAE (Masked Autoencoder) [20]:**
  - *Architecture*: Asymmetric ViT-B/16 with 75% patch masking
  - *Pretraining Objective*: Self-supervised, pixel reconstruction of masked image regions
  - *Training Data*: ImageNet-1k (1.28M images)
  - *Key Strengths*: Excels at texture/structure recovery; provides complementary low-level features to DINO's semantics.

- **CLIP (Contrastive Language-Image Pretraining) [48]:**
  - *Architecture*: ViT-B/16 image encoder (text branch disabled)
  - *Pretraining Objective*: Contrastive alignment of 400M image-text pairs, language is used as weak-supervision signal
  - *Training Data*: Web-crawled multimodal corpus
  - *Key Strengths*: Cross-modal concept grounding; robust to distribution shifts

Our experiments use frozen foundation models without fine-tuning: ViT-B/16 variants of DINO, MAE, and CLIP with 14×14 patch size and 224×224 input resolution. The CLIP text branch is excluded. For SAM, we use only the image encoder, downsampling positional embeddings to align spatial resolution with other models; the decoder is omitted as our focus is solely on representation learning. The specific hyperparameters of other modules are included in Table 8. We follow DINOSAUR [52] in setting the slot count for COCO and PASCAL to 6 slots, and for other datasets we use 7 slots. Both teacher and student branches are trained with the same number of slots. Using a teacher and a student with a different number of slots would be interesting to explore in future work.

## E   Dataset details

Here we extend details of all datasets used in this research:

*Abdominal dataset* is a public dataset from animal, phantom, and simulator abdominal surgeries [72]. We utilize 739260 frames for training, and 3000 frames with segmentation masks evaluation.

*Cholec dataset* [58] consists of 80 laparoscopic cholecystectomy videos. Following [35], we use 15,000 frames for training. 8,000 frames with segmentation annotations for evaluation.

*Thoracic dataset* includes data from 40 robot-assisted right upper lobectomies (RULs) for lung cancer, performed at Toronto General Hospital between 2014 and 2023. We use a total of 51,900 images for training and 800 manually annotated frames for evaluation.[2]

*COCO* (Common Objects in Context) [36] is a widely used benchmark for object detection, segmentation, and image captioning, consisting of 80 object categories. We use the 2017 split, with 118,000 images for training and 5,000 for validation.

*PASCAL* VOC 2012 [13] provides 11,530 images with segmentation masks for 20 object categories. Following standard protocol, we use 10,582 images for training and 1,449 for validation.

---

[2]An open-access version of this proprietary dataset is integrated as part of the federated learning benchmark in this work. See: https://github.com/PCASOlab/FORLA

Table 8: Hyperparameters of different network components.

| | Name | Type | Model file size | Feature dim |
|---|---|---|---|---|
| Foundation models | DINO | ViT-B/16 | 346 MB | 768 |
| | SAM | ViT-B/16 | 375 MB | 256 |
| | MAE | ViT-B/16 | 327 MB | 768 |
| | CLIP(image encoder) | ViT-B/16 | 437 MB | 768 |
| | Type | Input dim | Model file size | Feature dim |
| Adapters | MLP | 2560 | 56MB | 256 |
| | MoE | 2560 | 107MB | 256 |
| | AFM | 2560 | 158MB | 256 |
| | Slot num | Slot dim | Model file size | Iteration |
| Slot attention | 6 / 7 | 256 | 2.3 MB | 3 |
| | MLP hidden dim | Input dim | Model file size | Output dim |
| Teacher decoder | 1024 | 256 | 13M | 262 / 263 |
| Student decoder | 1024 | 256 | 23M | 2566 / 2567 |

*YTVIS* (YouTube-VIS) [66] is a benchmark for video instance segmentation, containing 8,858 videos spanning 40 object categories, with pixel-level masks across frames. We train on the 2,985-video training split from the 2021 version (78,810 frames) and evaluate on 4,210 validation frames.

*YTOBJ* (YouTube-Objects) [47] consists of 126 YouTube videos across 10 object categories, annotated with sparse bounding boxes for object tracking. We extract 388,050 frames from 100 videos for training and evaluate on 9,000 frames from 26 held-out videos.

# F  Additional Implementation Details

## F.1  Additional Details on Metrics

We evaluate our approach based on the quality of the slot attention masks using four primary metrics: Foreground Adjusted Rand Index (FG-ARI) [17], Mean Best Overlap (mBO) [46], Mean Best Hausdorff Distance (mBHD), and CorLoc [3]. FG-ARI, widely adopted in object-centric research, measures the similarity between predicted object masks and ground truth segmentation, specifically focusing on foreground regions. mBO evaluates the overlap between predicted and ground truth masks using intersection-over-union (IoU). It computes the average IoU after Hungarian matching between each ground truth and its best-matching predicted mask. While FG-ARI emphasizes segmentation quality regardless the permutation, mBO offers a broader assessment by channel matched IoU. To further assess spatial precision, we compute mBHD, which captures boundary-level accuracy and is particularly important for clinical applications. CorLoc measures localization accuracy by counting predicted object instances whose bounding boxes achieve IoU > 0.5 with a ground truth object.

## F.2  Additional Details on Experiment setup

Our experiments are conducted under three training regimes: (i) individual training, where each dataset is trained independently; (ii) centralized mixed training, where data from multiple datasets (surgical or natural) are pooled; and (iii) federated training, where each dataset is treated as a separate client without data sharing.

As each dataset can be considered as residing on a single client, we have a maximum of seven clients corresponding to seven distinct sub-domains. All images are resized as 224×224 for input (14×14 patches after ViT-B/16). For SAM, only the image encoder is used; positional embeddings are down-sampled to match the spatial resolution of the other models, and the decoder is omitted since representation learning is our focus. All adapter variants reduce the input feature dimensionality to 256 and use a slot embedding size of 256, following common practice in recent slot attention models [52, 62]. Each federated client is trained for a minimum of 100 epochs, with early stopping triggered if the student's reconstruction loss does not improve over 30 consecutive epochs. This stopping criterion was consistently met, and no late convergence was observed. The teacher's reconstruction loss is not used for early stopping, as it tracks a dynamic and more fluctuating target. Individual and

Table 9: Performance on surgical data using single foundation model's raw features for slot attention.

| Model | Abdominal | | | | Cholec | | | | Thoracic | | | |
|---|---|---|---|---|---|---|---|---|---|---|---|---|
| | mBO | mHD↓ | FG-ARI | CorLoc | mBO | mHD↓ | FG-ARI | CorLoc | mBO | mHD↓ | FG-ARI | CorLoc |
| DINO | **47.33** | **51.497** | **57.87** | 52.38 | **28.7** | **62.13** | **38.9** | 30.4 | 30.96 | 112.324 | 19.3 | 30.45 |
| SAM | 47.0 | 55.84 | 53.7 | **56.2** | 25.75 | 79.43 | 32.62 | **31.81** | **50.28** | **71.94** | **41.14** | **54.88** |
| MAE | 35.0 | 71.17 | 42.6 | 34.0 | 14.37 | 107.21 | 19.23 | 18.7 | 35.08 | 100.57 | 36.35 | 26.53 |
| CLIP | 23.0 | 75.14 | 28.9 | 19.8 | 10.9 | 97.26 | 13.41 | 10.39 | 14.99 | 117.26 | 8.71 | 11.75 |

centralized training baselines are run for 130 epochs. Our experiments were conducted using four NVIDIA RTX 6000 GPUs, with some GPUs assigned multiple clients. Each running client consumes approximately 6 GB of GPU memory after feature caching. Feature caching accelerates training by a factor of 10–20 for each client. A complete run of federated learning across all mixed sub-domains takes approximately 12 hours.

Centralized training on 1.4 million data points (frames) across 7 datasets with 130 epochs and batch size 16 takes 21.6 hours which is 1.8 times slower than FL. As the data can not be distributed across too many clients since unsupervised object centric learning needs a certain amount of data to discover meaningful concept, it would indeed be interesting to explore in more detail how one can achieve an optimal speedup/performance ratio in a FL training regime.

For the Natural image datasets we used optimal slot counts reported in the literature [11, 52]. For the surgical datasets we tuned the slot count to achieve a best score. We've also used some evidence from training SA models with both types of data [34, 33]. Choosing an optimal slot number is indeed important for slot attention algorithms. Our method is also fully compatible with SA approaches that use an adaptive slot count, but we decided to use more traditional approaches in this work for easier evaluation.

# G    Additional results

In this section, we extend experiments on using single or different combination of frozen/dapted foundation models, distillation dynamics on different dataset using different adapters, evaluation the performance of teacher branch (student is reported in the main paper), Comparison on Zero-shot transfer from natural to surgical domain, FORLA inference on videos when compared to DINO and SAM backbone, and more qualitative demonstration of feature adaptation and slot attention on different domains.

## G.1    Comprehensive Foundation Model Benchmarking

Tables 9 and 10 provide a detailed benchmark of four foundation models across surgical and natural domains. In this experiment we use only frozen features and no adapter or fine-tuning is applied. Each model is train on a single dataset individually. Three key insights emerge: 1) **Specialization-utility tradeoff:** DINO's self-supervised features excel on Cholec instruments (28.7 mBO) and COCO (23.96 mBO), indicating strong general object semantics. SAM with its segmentation-focused pretrained features achieves 50.28 mBO on Thoracic data (Table 9), outperforming DINO's 30.96 mBO, demonstrating out of domain generalization advantages. Reconstruction-based MAE underperforms on surgical data (-17.2 mBO vs. SAM) but shows unexpected competence on YTVIS (20.2 mBO) which could due to YTVIS requiring less high-level semantic features. 2) **Modality and domain mismatch:** CLIP's text-image alignment provides limited value for surgical domains (14.99 mBO Thoracic), suggesting medical imaging diverges from its web-scale pretraining. MAE trained on natural domain also transfers poorly to surgical scenes. 3) **Complementary Strengths:** SAM achieves highest CorLoc (56.2) while DINO leads in FG-ARI (57.87) on Abdominal data, and no single model dominates all metric on all data.

This analysis confirms that foundation models exhibit specialized capabilities aligned with their pre-training objectives and training data. FORLA's feature integration strategy (Eq. 1) allows synergistic combination of these complementary representations.

Table 10: Performance on natural data using single foundation model's raw features for slot attention.

| Model | COCO | | | | PASCAL | | | | YTVIS | | | | YTOBJ | |
|---|---|---|---|---|---|---|---|---|---|---|---|---|---|---|
| | mBO | mHD↓ | FG-ARI | CorLoc | mBO | mHD↓ | FG-ARI | CorLoc | mBO | mHD↓ | FG-ARI | CorLoc | mBO | CorLoc |
| DINO | **23.96** | **81.929** | **29.2** | 20.13 | 34.79 | 78.853 | 35.17 | 52.31 | **33.09** | **68.238** | **36.64** | **54.02** | 42.77 | 54.78 |
| SAM | 22.61 | 93.192 | 25.02 | **20.5** | **36.98** | **73.992** | **35.59** | **56.04** | 32.62 | 68.704 | 35.89 | 53.66 | **42.86** | **54.95** |
| MAE | 17.08 | 101.592 | 18.83 | 8.59 | 25.95 | 91.41 | 22.93 | 31.87 | 20.2 | 87.701 | 20.81 | 15.94 | 33.86 | 29.91 |
| CLIP | 12.73 | 109.433 | 14.14 | 3.46 | 14.52 | 109.054 | 9.82 | 27.47 | 14.37 | 107.217 | 19.23 | 23.15 | 22.27 | 9.02 |

Table 11: Adaptation of features from a single foundation model on Abdominal, YTVIS and YTOBJ.

| Adapter | Model | Abdominal | | | | YTVIS | | | | YTOBJ | |
|---|---|---|---|---|---|---|---|---|---|---|---|
| | | mBO | mHD↓ | FG-ARI | CorLoc | mBO | mHD↓ | FG-ARI | CorLoc | mBO | CorLoc |
| MLP | Concat | _51.85_ | _43.025_ | _59.04_ | _69.25_ | 35.16 | 67.22 | 38.25 | 58.08 | _48.62_ | _60.31_ |
| | DINO | 45.35 | 53.653 | 50.55 | **63.6** | 36.42 | 66.584 | **40.91** | 62.13 | **46.10** | **60.02** |
| | SAM | 41.78 | 69.091 | 46.08 | 57.72 | 35.93 | **64.650** | 39.61 | **63.16** | 37.72 | 51.92 |
| | MAE | 40.84 | 59.286 | 47.28 | 52.3 | 21.94 | 83.329 | 22.97 | 20.67 | 33.52 | 26.83 |
| AFM | Concat | _50.34_ | _44.398_ | _59.04_ | _70.73_ | 33.94 | 66.637 | 37.54 | 54.59 | _48.98_ | _65.11_ |
| | DINO | 41.99 | 63.738 | 47.46 | 56.73 | **36.52** | 65.752 | **41.01** | 61.18 | **48.17** | **62.00** |
| | SAM | 41.73 | 66.985 | 46.05 | 56.65 | 36.06 | **64.939** | 39.51 | **64.04** | 41.09 | 51.35 |
| | MAE | **42.85** | **59.433** | **49.70** | 54.37 | 21.20 | 85.221 | 22.66 | 19.36 | 34.87 | 27.58 |

Table 12: Evaluation of Frozen and Adapted models using 3 vs. 4 foundation models. FM: Foundation Model, Surgical:Abdominal, Natural:YTOBJ.

| Setting | FM number | Surgical | | | Natural | | |
|---|---|---|---|---|---|---|---|
| | | mBO | FG-ARI | Cor-Loc | mBO | FG-ARI | Cor-Loc |
| Frozen | 3 | 39.78 | 51.28 | 54.26 | 42.92 | 33.74 | _51.85_ |
| | 4 | _48.65_ | _54.07_ | _64.30_ | _43.33_ | _38.78_ | 50.56 |
| Adapted | 3 | 49.58 | 58.52 | 69.58 | 47.32 | 41.86 | 58.97 |
| | 4 | **50.34** | **59.04** | **70.73** | **48.98** | **45.32** | **65.11** |

## G.2 Efficacy of Adaptation Layer for Single Foundation Models

Next, to assess the benefit of feature adaptation, we evaluate the performance when using a single foundation model augmented with a lightweight adapter module. In these experiments, we attach either a simple **MLP** adapter or the proposed **AFM** adapter on top of the frozen foundation model, then train the adapter (keeping the foundation model weights fixed) for the downstream object-discovery. Here MOE is not used as it is only applicable for multiple foundation models. Table 11 (middle) reveals the performance:All foundation model show improvement with both adapter on YTVIS data, particularly AFM boost SAM's CorLoc from 53.66 to 64.65. Adapters recover 54% of MAE's performance gap vs. SAM on abdominal data (35.0 to 42.8 mBO). Importantly, the performance of foundation models when used individually lags the performance of concatenated adapted baselines, in particular when applied to a domain that is new to foundation models (+5 mBO, + 9 FG-ARI, +9 Cor-Loc).

## G.3 Using Different Numbers of Foundation Models

We further examine the impact of using different numbers of foundation models within FORLA. When employing four foundation models (including ViT-B/16, the smaller ViT variant), the computational overhead increases modestly from 1.4 ms (using only DINO) to 3.8 ms per image. Considering recent domain-specific foundation models such as RADIOv2.5 [21], one may question whether fewer models could provide sufficient representational power, particularly in the surgical domain.

To investigate this, we evaluated configurations using either three (DINO, SAM, and CLIP) or four foundation models, under both frozen and adapted settings. As shown in Table 12, the configuration with four foundation models consistently achieved the best performance, especially on surgical images, which are typically underrepresented in the pretraining of generic foundation models. Despite the modest increase in computational cost (an additional 2.4 ms per image), the performance gains justify the use of four foundation models for achieving stronger generalization and robustness.

Table 13: Additional results of self-distillation on the single data with MLP and MOE adapters.

| Adapter | Method | Abdominal | | | | Thoracic | | | |
|---|---|---|---|---|---|---|---|---|---|
| | | mBO | mHD↓ | FG-ARI | CorLoc | mBO | mHD↓ | FG-ARI | CorLoc |
| MLP | w/o self-ditill | 51.85 | 43.025 | 59.04 | 69.25 | 54.00 | 72.358 | 43.59 | 58.00 |
| | w self-distill | 56.06 | 35.005 | 63.09 | 79.23 | 44.86 | 85.707 | 36.73 | 49.28 |
| MOE | w/o self-ditill | 51.83 | 42.758 | 57.54 | 72.02 | 51.10 | 74.670 | 41.87 | 55.89 |
| | w self-distill | **57.05** | **35.039** | **63.27** | **79.62** | **62.16** | **52.544** | **47.96** | **72.32** |

Table 14: Decoder performance of the teacher branch in comparison to student branch in FORLA.

| Domain | Sub-domain | mBO | | mHD ↓ | | FG-ARI | | CorLoc | |
|---|---|---|---|---|---|---|---|---|---|
| | | *Teacher* | *Student* | *Teacher* | *Student* | *Teacher* | *Student* | *Teacher* | *Student* |
| Surgical | Abdominal | 51.29 | 57.86 | 38.458 | 34.371 | 58.31 | 64.88 | 77.73 | 80.30 |
| | Cholec | 32.52 | 34.20 | 57.247 | 56.942 | 42.31 | 44.35 | 52.13 | 54.49 |
| | Thoracic | 56.86 | 61.86 | 52.702 | 50.771 | 43.60 | 47.58 | 76.04 | 75.42 |
| | Average | 46.89 | 51.31 | 49.469 | 47.36 | 48.07 | 52.27 | 68.63 | 70.07 |
| Natural | COCO | 26.72 | 27.22 | 93.473 | 93.541 | 28.41 | 30.11 | 66.88 | 64.94 |
| | PASCAL | 39.36 | 40.83 | 76.277 | 75.642 | 36.41 | 39.51 | 65.38 | 64.84 |
| | YTVIS | 37.96 | 38.51 | 61.396 | 62.472 | 41.63 | 43.08 | 65.50 | 64.92 |
| | YTOBJ | 51.06 | 51.92 | – | – | – | – | 62.03 | 66.87 |
| | Average | 38.78 | 39.62 | 77.05 | 77.22 | 35.48 | 37.57 | 64.95 | 65.39 |

## G.4 Distillation Dynamics

Table 13 (bottom) shows additional effects of self-distillation on abdominal and thoracic data with MLP and MOE adapter: 1) **MLP overfitting:** Self-distillation improves Abdominal CorLoc (improved performance score by 10) but harms Thoracic performance (-8.7 points performance), suggesting MLP could over-fit to smaller specialized dataset with less constrains for feature reconstruction. 2) **MOE Robustness:** MOE adapters gain +16.4 CorLoc on Thoracic with distillation, leveraging expert gates to preserve generalizability. This once again confirmed that MLP could be the least suitable adapter choice in our FORLA federated object-centric learning framework, as demonstrated in Table 2.

## G.5 Teacher Decoder Analysis

In FORLA, teacher and student branch share the same global adapter and Slot attention module, while having different decoders for reconstructing features and slot attention masks. We analyzed the performance of the decoder branches by comparing the teacher and student models across all datasets. As shown in Table 14, the teacher branch performance is consistently competitive and often close to the performance of the student decoder, despite the fact that its adapter is not directly optimized but rather updated through EMA (early stage) or local FedAvg (later stage).

Table 14 reveals nuanced performance differences between teacher and student decoders in federated learning: 1) **Dominance on surgical domain:** Student decoder achieves +4.42 average mBO improvement (51.31 vs. 46.89), demonstrating superior object discovery from federated knowledge aggregation; it maintains boundary precision with 4.7% lower mHD (47.36 vs. 49.47), crucial for anatomical structures; 2.44 FG-ARI gain highlights better foreground-background separation. 2) **Gains on natural domain:** Student leads marginally in mBO (+0.84) FG-ARI (+2.09), and CorLoc (0.44). This validates the effectiveness of our teacher-student design, where the student benefits from both local and global knowledge transfer via FL and reconstruct more constrained features, while the teacher adapts more aggressively and encourages the student to re-discover more specialized and transferable object-centric representations.

## G.6 Compared to Zero-shot transfer performance of slot attention

We performed additional testing that confirmed our hypothesis which was that, if the domain gap is small, the zero-shot and transfer learning will guarantee good performance. However, FORLA FL will outperform transfer learning when the domain gap is large (Table 15). We first tested zero-shot transfer of slot attention trained on Natural images (PASCAL and YTOBJ) to surgical images (Abdominal). In this case zeroshot performance is significantly lower compared to FORLA and even compared to models individually trained on abdominal data.

Table 15: Comparison of zero-shot transfer from Natural to surgical, individual training, and FORLA.

| Method | mBO | FG-ARI | Cor-Loc |
|--------|-----|--------|---------|
| Zero-shot (PASCAL → Abdominal) | 35.61 | 39.81 | 35.90 |
| Zero-shot (YTOBJ → Abdominal) | 33.02 | 37.69 | 30.65 |
| Individual | 50.34 | 59.04 | 70.73 |
| FORLA | **57.86** | **64.88** | **80.30** |

### G.7 Inference on Videos

We demonstrate that a slot attention module trained with our FORLA framework can be directly applied to video scene decomposition by leveraging RNN-based slot inference techniques [67]. Specifically, we evaluated our FORLA-trained slot attention model on the YTOBJ video dataset sampled at 1 frame per second (fps). For comparison, we trained individualized slot attention (SA) models directly on YTOBJ using adapted versions of foundation models, including DINO (equivalent to DINOSAUR [52]) and SAM.

As shown in Figures 4 and 5, the slots produced by FORLA maintain strong temporal consistency across different video sequences. The performance of individualized models reveals that SAM- and DINO-based SA models exhibit distinct strengths: the SAM-based model excels at decomposing close-up scenes such as animals or vehicles captured by a near camera, while the DINO-based model performs better in delineating objects from the background in distant or wide-angle scenes.

FORLA however consistently tracks objects both in near and far scenes in a more fine-grained and semantically meaningful manner. Unlike the SAM or DINO-based models, FORLA is less likely to segment objects into implausible parts (e.g., splitting a cat or car into non-semantic regions), demonstrating stronger object-level coherence and generalization.

## H Visualization of Feature Representation

Figures 6 and 7 provide additional visualizations of PCA maps and Slot Attention (SA) masks for surgical and natural image domains, respectively, across different methods. We visualize the first three principal components of the feature representations using RGB channels. These include both frozen features from foundation models and adapted features learned through our FORLA federated learning framework.

In the surgical domain, the frozen foundation model features demonstrate a general understanding of texture-based separation—for example, surgical instruments and tissues often appear as different colors in the PCA map. This indicates some level of semantic separation. However, the foundation features struggle to distinguish between multiple instances of similar instruments, as they are often represented with the same color. In contrast, the FORLA-adapted features effectively assign distinct semantic representations to different instrument instances, enabling clearer separation. Additionally, FORLA representations offer greater differentiation between various tissue textures. This is reflected in the PCA maps, where different tissues are represented by homogenous yet distinct colors, aiding the SA module in grouping regions into semantically meaningful categories.

In the natural image domain, foundation model features can delineate some salient objects, due to their pretraining on natural image datasets, but the representations remain relatively coarse. With FORLA's unsupervised adaptation, the feature representations become significantly more fine-grained. For example, tiny objects that are otherwise overlooked in foundation features become clearly highlighted in the PCA maps. The model also learns to extract subtle cues from cluttered backgrounds, such as the texture of plants, and can even distinguish between visually similar objects located close to each other. These properties of the adapted representation enable the downstream SA module to decompose scenes into semantically coherent segments.

Such fine-grained representation and semantically meaningful decomposition suggests that FORLA-learned representations can potentially generalize well to downstream tasks, including semi-supervised or weakly supervised segmentation, as well as a variety of prediction tasks that benefit from pre-segmented scene understanding [54].

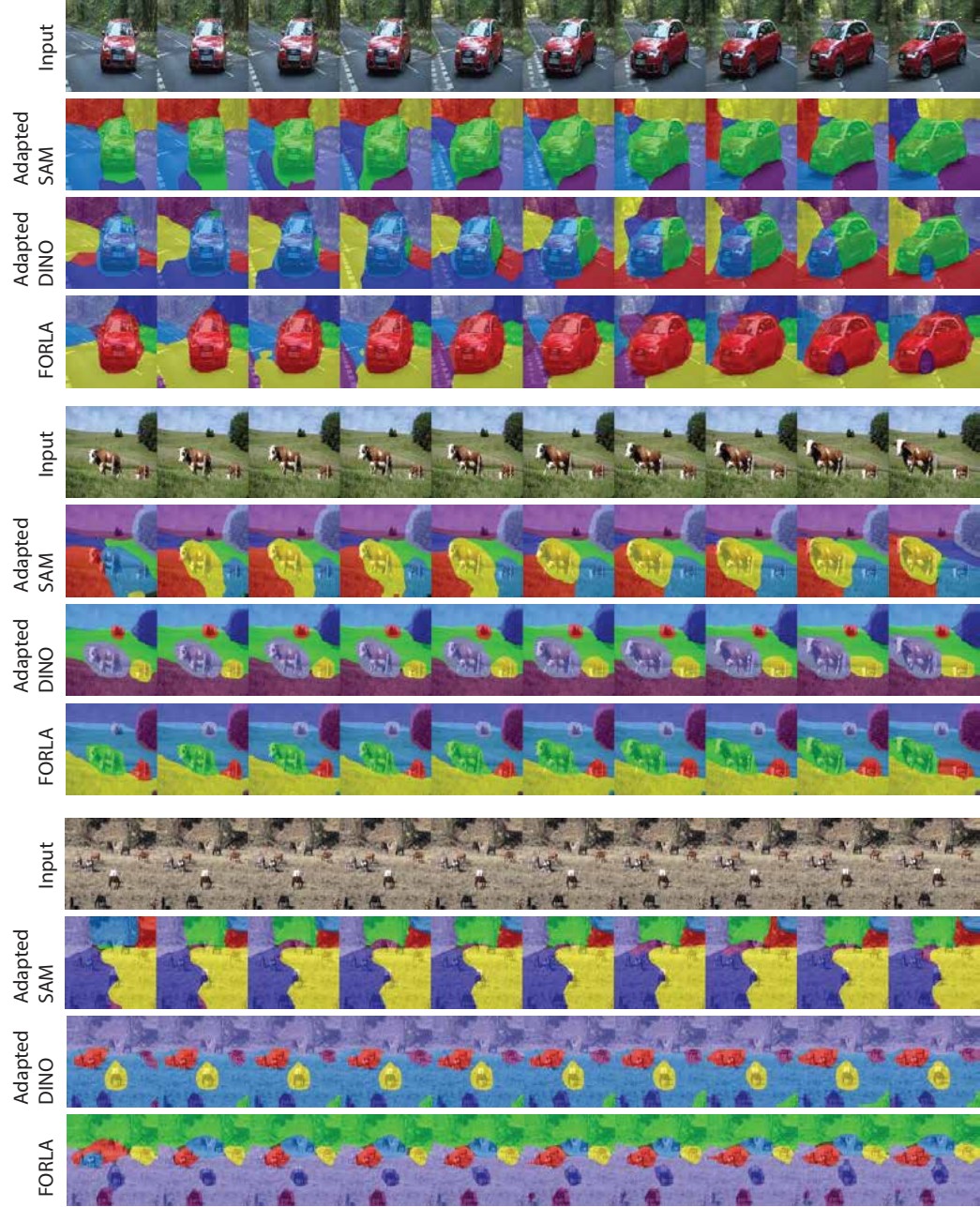

Figure 4: Inference using RNN like slot initialization [67] on YTOBJ videos. We compared to individualized trained SA models on YTOBJ using adapted single foundation model including DINO (as used in DINOSAUR [52]) and SAM.

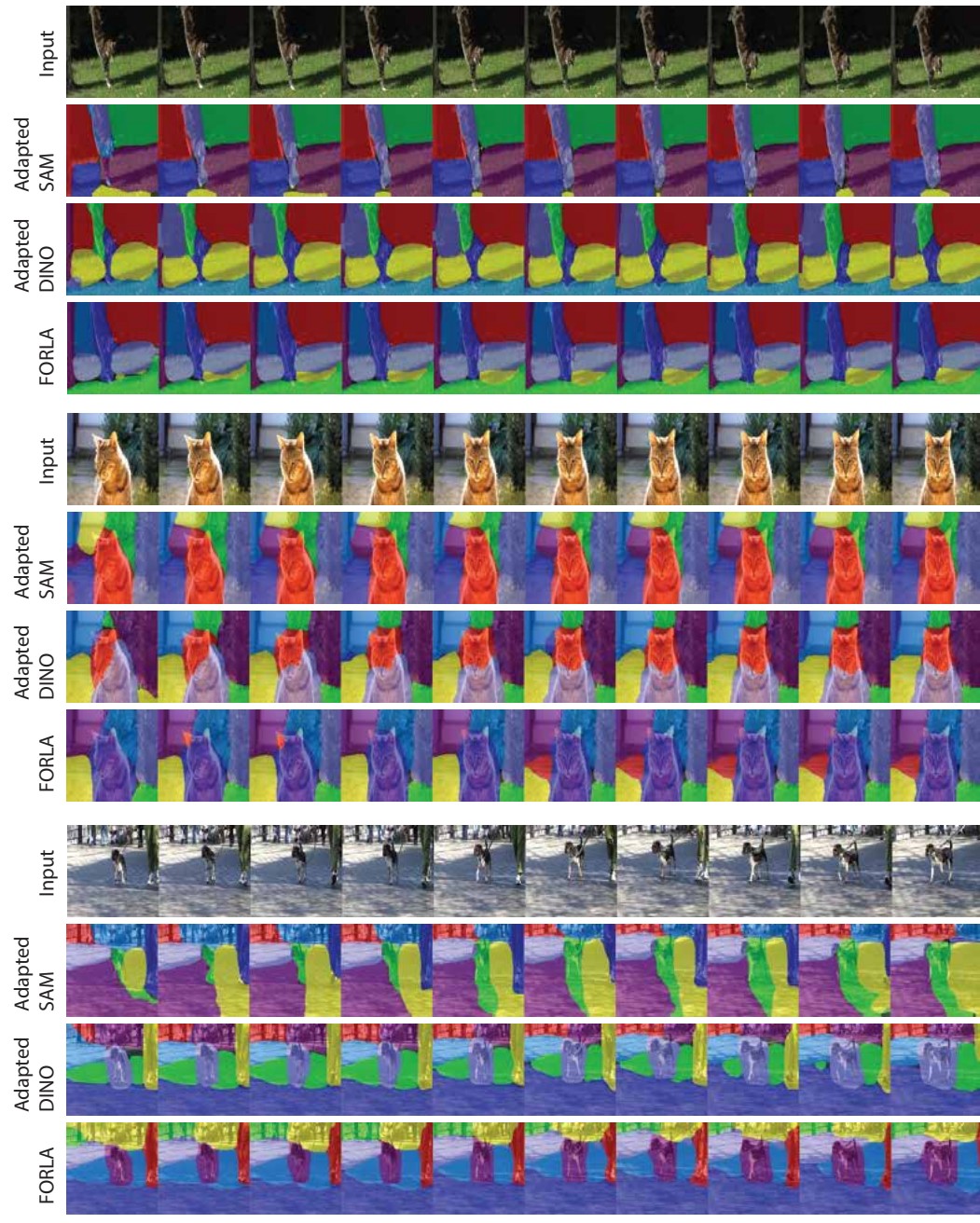

Figure 5: Additional results on YTOBJ videos compared to individualized trained SA models with single foundation model adaptation.

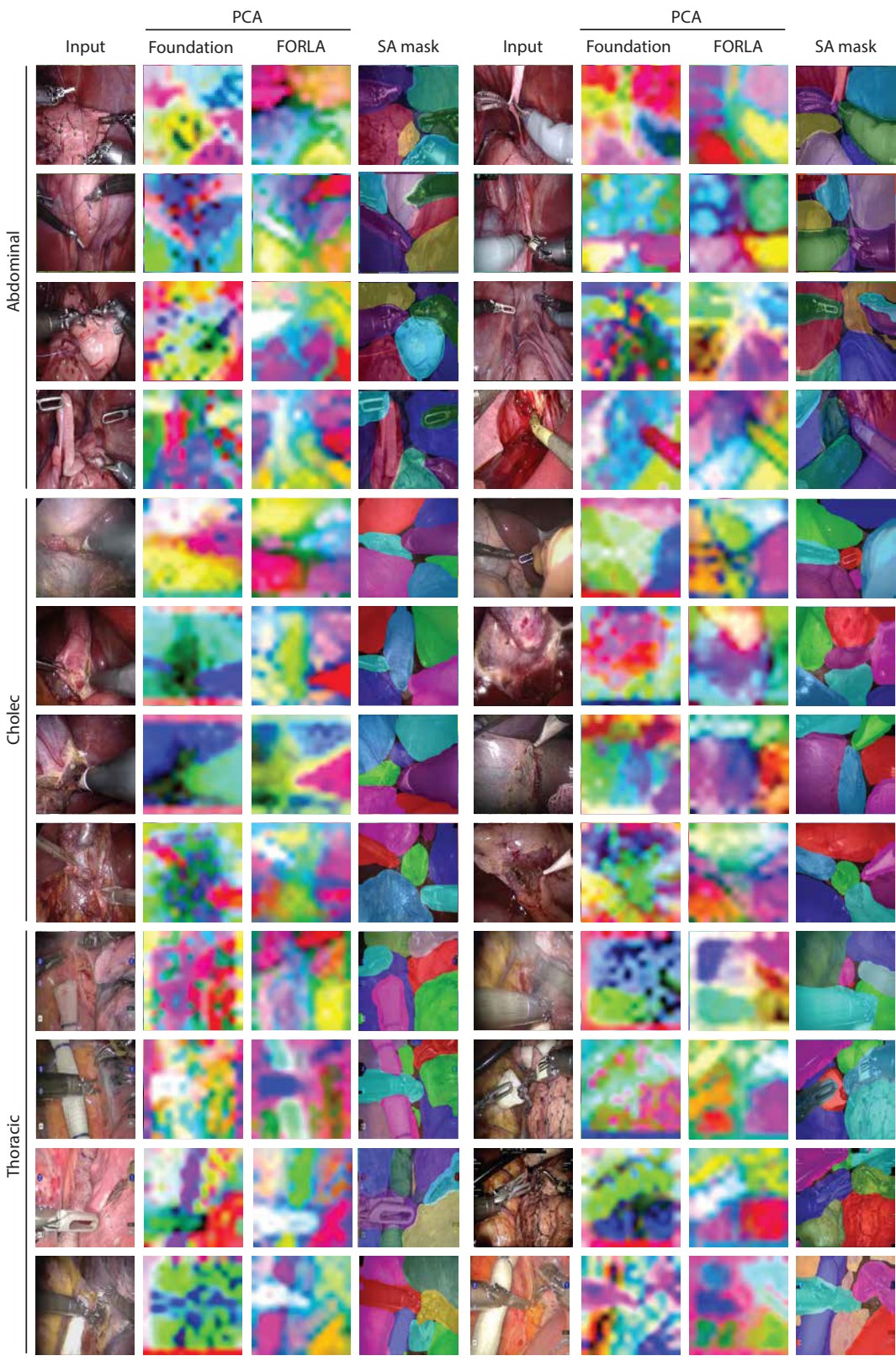

Figure 6: PCA map of raw foundation model features and FORLA representation on surgical image samples. Slot attention (SA) masks of FORLA are shown at 4th and 8th columns.

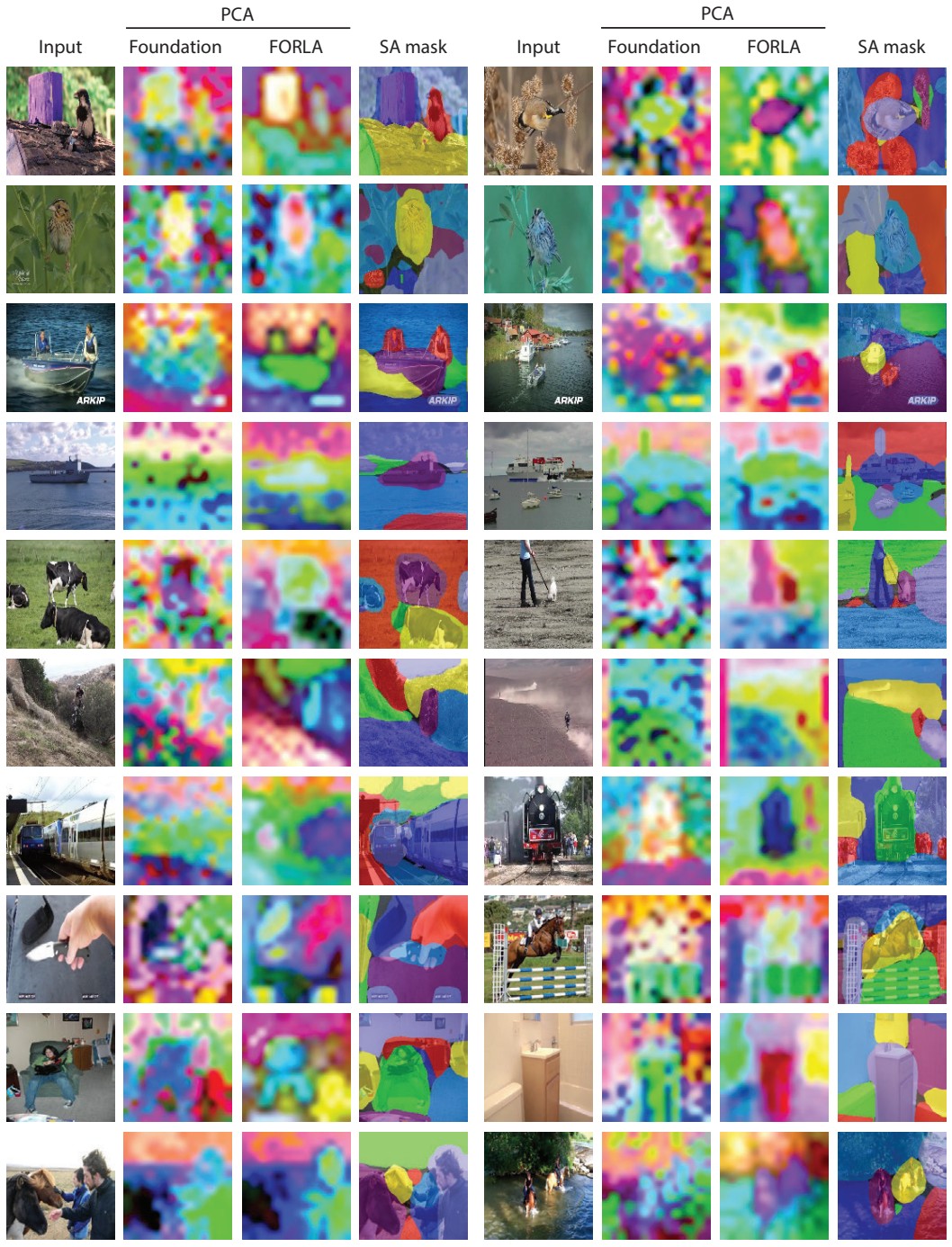

Figure 7: PCA map of raw foundation model features and FORLA representation on natural image samples. Slot attention (SA) masks of FORLA are shown at 4th and 8th columns.

