# OpenReview forum: "FORLA: Federated Object-Centric Representation Learning with Slot Attention"
_NeurIPS.cc/2025/Conference — NeurIPS 2025 poster_

### Official Review · Reviewer_dX9d · 2025-07-01

**Clarity:** 2
**Significance:** 2
**Originality:** 3
**Rating:** 4
**Confidence:** 4

**Summary:**

The paper addresses an interesting question of efficiently learning object-centric features from datasets in various domains. In particular paper proposed to use adapters for each domain that transform general features from a combination of foundational models with a sophisticated way of training each adapter on the client dataset, while using a shared SA module.

**Questions:**

Please address the weaknesses of the paper.

Additional questions:
1. YTOBJ is a storage choice for the multi-object scene decomposition. If I remember correctly, most videos consist of objects and background. Thus, other video domains, such as robotics videos from DROID, could be a much better choice there.
2. FT-DINOSAUR showed that it is a significant zero-shot transfer is possible from the COCO natural image domain to other domains. Thus zero-shot baseline could be interesting to see how much such training is needed.
3. How is the number of slots for each dataset determined? Is the whole pipeline sensitive to it?

**Ethical Concerns:**

["NO or VERY MINOR ethics concerns only"]

**Final Justification:**

I appreciate the effort made during the rebuttal.
As authors are planning to simplify the presentation and FLORA works better SAM masks (I think this should be highlighted and discussed to show the difference between OCL methods and zero-shot segmentation), I increased my score to weak accept.

**Limitations:**

Yes

**Paper Formatting Concerns:**

No problems

**Quality:**

3

**Strengths And Weaknesses:**

### Strengths:

1. Overall, the paper studies the important problem of how to combine representations from general foundational models with different domains effectively.
2. I think the paper is one of the first to show how OCL could be applied to different domains simultaneously, showing the importance of using dataset structure (e.g., that different sources of data share some features, but still have different biases).
3. The paper studies many research questions during its analysis. The future work and limitations

### Weaknesses:

1. (Clarity) The proposed approach is a quite complex combination of several topics that are not trivially connected, thus making the presentation challenging. It would be great if the authors would make emphasize to making take-away messages and reshape the results accordingly. For example, different ways to adapt features are not novel, and while an important analysis of the better choice could be presented in the appendix, if this is not the message of the paper.
2. The paper states that it is doing Representation Learning; however, only masks (e.g., object discovery) are evaluated. It would be great to include other ways of evaluation representations (class label or attribute prediction). It would be great to see if learned features in such an involved way are indeed more informative other the domain than the naive average training. Also, while SAM is used only for features, it is not clear if the (supervised) masks from the SAM decoder are much better already than those obtained from decomposition. Thus, such zero-shot baseline should also be present.
3. Use of several foundational models is potentially quite heavy during inference. It would be great to show how much we are losing in inference speed because of such a combination. Could Agglomerative models like RADIOv2.5 [4] (combined DINO, SAM, and CLIP) be enough already?
4. Some recent related work is potentially missing. For example, SlotContrast [1] and CTRL-O [2], showed that adaptation of dense features with an MLP is helping with the contrastive objective in videos and vision-language accordingly, while SPOT [3] was using student-teacher distillation for images to improve mask quality.

[1] Temporally Consistent Object-Centric Learning by Contrasting Slots, CVPR 2025
[2] CTRL-O: Language-Controllable Object-Centric Visual Representation Learning, CVPR 2025
[3] SPOT: Self-Training with Patch-Order Permutation for Object-Centric Learning with Autoregressive Transformers, CVPR 2024
[4] RADIOv2.5: Improved Baselines for Agglomerative Vision Foundation Models,  CVPR 2025

---

> ### Author Rebuttal · Authors · 2025-07-30
>
> We appreciate the reviewer’s recognition that our paper is among the first to apply OCL across multiple domains and demonstrate the benefits of leveraging dataset structure via federated learning. We will restructure the results to emphasize key takeaways, expand the evaluation metrics, and address missing baselines, efficiency concerns, and sensitivity analyses as suggested.
>
> ---
>
> **Q:** *The proposed approach is a quite complex combination of several topics that are not trivially connected, thus making the presentation challenging. It would be great if the authors would make emphasize to making take-away messages and reshape the results accordingly. For example, different ways to adapt features are not novel, and while an important analysis of the better choice could be presented in the appendix, if this is not the message of the paper.*
>
> **A:** We agree that the clarity of the paper can be further improved, which is also in line with comments from other reviewers. In the revised version we will move content that is not the core contribution, including the description of adapter designs, performance and benchmarks with a single foundation model, and detailed datasets description to the Appendix. The key message we want to highlight is that FORLA surpasses centralized slot attention for multi-domain data and show how the network can exploit local structural coherence while learning a global representation.
>
> ---
>
> **Q:** *The paper states that it is doing Representation Learning; however, only masks (e.g., object discovery) are evaluated. It would be great to include other ways of evaluation representations (class label or attribute prediction). It would be great to see if learned features in such an involved way are indeed more informative other the domain than the naive average training.*
>
> **A:** Testing the learned representation on tasks like classification or attribute prediction is definitely valuable and interesting and is part of our ongoing work. We followed other object centric representation papers (e.g MONET, IODINE and FT-DINO) focusing on the object segmentation masks that provide fine-grained evaluation metrics to compare different approaches.
>
> 1. Klaus Greff, et al.,Multi-Object Representation Learning with Iterative Variational Inference  2. Christopher P. Burgess, et al., MONet.
>
> ---
>
> **Q:** *while SAM is used only for features, it is not clear if the (supervised) masks from the SAM decoder are much better already than those obtained from decomposition. Thus, such zero-shot baseline should also be present.*
>
> **A:** We perform zero-shot evaluation of full SAM models (with ViT-b and the ViT-h). We found SAM ViT-h is able to surpass slot attention models trained with FORLA on the natural images but fails on the very specific surgical domain (Thoracic). The underlying reason could be that the thoracic data is not publicly available and not included in the training data of SAM. The inference time for SAM ViT-h of one image is 1.1s (0.85s for ViT-b), while the slot attention model takes only 3.75ms. Another advantage of OCL over SAM is that training SAM on new domains would require mask supervision and, most importantly, SAM will not capture the objectness (over decompose object parts to tiny masks) unless instructed to do so via annotation.
>
> |          |       |     **PASCAL**    |         | | **Abdominal** |         |  | **Thoracic (+)** |         |
> |----------|------------|---------|---------|---------------|---------|---------|--------------|---------|---------|
> |  Method   | **mBO**    | **FG-ARI** | **Cor-Loc** | **mBO**       | **FG-ARI** | **Cor-Loc** | **mBO**      | **FG-ARI** | **Cor-Loc** |
> | **SAM (ViT-b)** | 23.79      | 24.55   | 35.16   | 45.76         | 52.10  | 39.15   | 18.10        | 16.41   | 8.35    |
> | **SAM (ViT-h)** | 54.90      | 57.31   | 79.12   | 66.66         | 69.81 | 65.47   | 33.57        | 30.48   | 23.73   |
> | **FORLA**      | 40.83      | 39.51   | 45.04   | 57.86         | 64.88 | 80.30   | 61.86        | 47.58   | 75.42   |
>
> *Performance comparison of SAM and FORLA.+Note that thoracic is less publically available*
>
> ---
>
> **Q:** *Use of several foundational models is potentially quite heavy during inference. It would be great to show how much we are losing in inference speed because of such a combination. Could Agglomerative models like RADIOv2.5 [4] (combined DINO, SAM, and CLIP) be enough already?*
>
> **A:** Using four foundation models (with VIT-b/16) increases the computation overhead from 1.4 ms (with only DINO) to 3.8 ms. In addition to testing performance with a single foundation model (already in Appendix), we tested a configuration with  3 foundation models (DINO, SAM,CLIP) under both frozen and adapted setting. As shown in the table below, we found that 4 foundation models achieve best performance, especially in surgical images which is mostly not included in the pretraining of foundation models. Nevertheless, RADIOv2.5 provides additional insight on combining foundation models and will be mentioned in the related work.
>
> |              |                          |  |      **Surgical**   |         | |  **Natural**   |         |
> |-------------|------------|-------------|---------|---------|--------------------|---------|---------|
> | **Setting** |  **FM number** | **mBO**                 | **FG-ARI** | **Cor-Loc** | **mBO**            | **FG-ARI** | **Cor-Loc** |
> | Frozen      | 3                      | 39.78                   | 51.28   | 54.26   | 42.92              | 33.74   | 51.85   |
> |             | 4                      | 48.65                   | 54.07   | 64.30   | 43.33              | 38.78   | 50.56   |
> | Adapted     | 3                      | 49.58                   | 58.52   | 69.58   | 47.32              | 41.86   | 58.97   |
> |             | 4                      | 50.34                   | 59.04   | 70.73   | 48.98              | 45.32   | 65.11   |
>
> *Evaluation of Frozen and Adapted models using 3 vs. 4 foundation models.FM: Foundation Model, Surgical:Abdominal, Natural:PASCAL*
>
> ---
>
> **Q:** *Some recent related work is potentially missing. For example, SlotContrast [1] and CTRL-O [2], showed that adaptation of dense features with an MLP is helping with the contrastive objective in videos and vision-language accordingly, while SPOT [3] was using student-teacher distillation for images to improve mask quality.*
>
> **A:** We appreciate the pointers to additional literature in this fast growing area, and we will add them to the related work section. A few contributions in these papers can be directly applied to our work in the future. For instance the loss in SlotContrast and the permutation of patches (removing of localization bias) in SPOT. For the CTRL-O, it would be interesting to include clients with language data as well.
>
> ---
>
> **Q:** *YTOBJ is a storage choice for the multi-object scene decomposition. If I remember correctly, most videos consist of objects and background. Thus, other video domains, such as robotics videos from DROID, could be a much better choice there.*
>
> **A:** Thank you for the recommendation of a new dataset. We chose YTOBJ as it partially overlaps with other Natural domains and lends well to the federated learning exploiting regularities in sub-domains. That being said, the robotic data is interesting for our problem as it lies in between a specialized dataset such as robotic surgery, and a Natural dataset with actuators acting on everyday objects. We will mention this in the discussion section and hopefully add this dataset to our future experiments.
>
> ---
>
> **Q:** *FT-DINOSAUR showed that it is a significant zero-shot transfer is possible from the COCO natural image domain to other domains. Thus zero-shot baseline could be interesting to see how much such training is needed.*
>
> **A:** Thank you for this question. We performed additional testing that confirmed our hypothesis which was that, if the domain gap is small, the zero-shot and transfer learning will guarantee good performance. However, FORLA FL will outperform transfer learning when the domain gap is large.We first tested zero-shot transfer of slot attention trained on Natural images (PASCAL and YTOBJ) to surgical images (Abdominal). In this case zeroshot performance is significantly lower compared to FORLA and even compared to models individually trained on abdominal data.
>
> | **Method**                          | **mBO** | **FG-ARI** | **Cor-Loc** |
> |-------------------------------------|---------|------------|-------------|
> | Zero-shot (PASCAL → Abdominal)      | 35.61   | 39.81      | 35.90       |
> | Zero-shot (YTOBJ → Abdominal)       | 33.02   | 37.69      | 30.65       |
> | Individual                          | 50.34   | 59.04      | 70.73       |
> | FORLA                               | 57.86   | 64.88      | 80.30       |
>
> *Comparison of zero-shot transfer, individual training, and FORLA*
>
> We additionally compare zero shot and transfer finetuning from COCO to PASCAL where the domain gap is smaller, and show that FORLA has an advantage in this setting, but with a smaller margin.
>
> | **Metric** | **Zeroshot** | **Transfer fine-tune** | **FORLA** |
> |------------|--------------|------------------------|-----------|
> | mBO        | 37.20        | 37.80                  | 40.80     |
>
> *Comparison of zeroshot,tranfer from COCO to PASCAL.*
>
> ---
>
> **Q:** *How is the number of slots for each dataset determined? Is the whole pipeline sensitive to it?*
>
> **A:** For the Natural image datasets we used optimal slot counts reported in the literature [10,43]. For the surgical datasets we tuned the slot count to achieve a best score. We've also used some evidence from training SA models with both types of data. Choosing an optimal slot number is indeed important for slot attention algorithms. Our method is also fully compatible with SA approaches that use an adaptive slot count, but we decided to use more traditional approaches in this work for easier evaluation.

---

> ### Author Response · Authors · 2025-08-06
>
> Dear Reviewer dX9d,
>
> Thank you once again for your valuable review. We appreciate your acknowledgment of having read our rebuttal and we would like to ask whether our response has adequately addressed all of your comments. If there are any reamaining concerns, we would like the chance to respond before the discussion period is over.
>
> Thanks again for your thoughtful review!
>
> Best regards,
>
> Authors

---

> > ### Comment · Reviewer_dX9d · 2025-08-06
> >
> > I appreciate the effort made during the rebuttal. As authors are planning to simplify the presentation and FLORA works better SAM masks (I think this should be highlighted and discussed to show the difference between OCL methods and zero-shot segmentation), I increased my score towards acceptance.

---

> ### Author Response · Authors · 2025-08-06
>
> Thank you very much for your response and for increasing your score! We are happy to incorporate the suggested improvements, including simplifying the presentation for clarity and highlighting how FLORA performs with SAM masks, especially in zero-shot settings across different domains. Thanks again for your constructive feedback.

---

### Official Review · Reviewer_mowD · 2025-07-02

**Clarity:** 3
**Significance:** 3
**Originality:** 2
**Rating:** 4
**Confidence:** 4

**Summary:**

This paper extends the object-centric representation learning from the centralized setting the distributed federated setting. Several techniques are assembled to enable this extension. First, a shared feature adapter is introduced and trained collaboratively across clients to adapt features from foundation models. Second, two-branch student–teacher architecture is introduced, a student decoder learns to reconstruct full features from foundation models, while a teacher decoder reconstructs their adapted, low-dimensional counterpart. Experiments show the effectiveness of the proposed method.

**Questions:**

See the weakness above.

**Ethical Concerns:**

["NO or VERY MINOR ethics concerns only"]

**Limitations:**

Yes.

**Paper Formatting Concerns:**

No.

**Quality:**

3

**Strengths And Weaknesses:**

Strengths
- The paper is well presented.
- The proposed method is technically sound.
- Experiments and analysis have done to verify the effectiveness of the proposed method.

Weakness
- The technical contribution is limited. The proposed method essentially ensembles several mature techniques to adapt the object centric representation learning problem from the conventional setting the federated learning setting.

- Adding adapters to finetune a pretrained model for the target task is common.  Feature reconstruction is also a widely used technique is self-supervised learning, and the knowledge distillation between sever model and client model is also mature technique in federated learning.

- Can the proposed method generalizes to other object centric representation learning approaches beyond slot attention? I did not find any  specific design for the slot attention. I guess other object centric representation learning approaches can also be employed and extended to the federated learning setting. It would be interesting to see these results as well.

---

> ### Author Rebuttal · Authors · 2025-07-30
>
> We thank the reviewer for highlighting the technical soundness and clear presentation of our work. We will  further emphasize the main technical result and novelty of representation learning that combines slot attention’s success in mixed domains with federated learning to exploit structural coherence in local datasets. We will also strengthen the discussion of our novel contributions beyond existing techniques and provide insight on how our approach generalizes to other OCL methods.
>
> ---
>
> **Q:** *The technical contribution is limited. The proposed method essentially ensembles several mature techniques to adapt the object centric representation learning problem from the conventional setting the federated learning setting.*
>
> **A:** We will clarify the novelty and contribution in the updated manuscript. Object-centric learning has not been scaled to large multi-domain data (e.g. including surgical and natural, or multiple domains of natural images) and ours is the first study of federated unsupervised object centric learning. As SA works by discovering repeated occurrence of objects in large datasets, disentangling the concepts is crucial and we show it can be achieved with FL. We are thus first to propose using FL to scale object centric learning and we clearly demonstrate how to implement representation disentanglement using a dual branch design, feature adaptation and knowledge distillation and why these approaches work.
>
> ---
>
> **Q:** *Adding adapters to finetune a pretrained model for the target task is common. Feature reconstruction is also a widely used technique is self-supervised learning, and the knowledge distillation between sever model and client model is also mature technique in federated learning.*
>
> **A:** Thank you for this comment. We acknowledge that adding adapters to a stack of pretrained models is not our core contribution. We tested several classic adapter designs to investigate FORLA's sensitivity to adapter type, and while the data did show it's important to carefully consider the right choice of the adapter (see also reply to reviewer TTbE regarding AFM performance), we will try to hit a right balance of detail and informativeness in the revision. We also agree with another reviewer to move the details of these designs to the Appendix while focusing more on our main contribution. As for knowledge distillation (KD), its use in slot attention is a technique that was proposed only recently and thus still requires specific and innovative solutions. For example, our method not only uses KD for FL (which is a mature technique) but it leverages FL to stabilize the loss for slot attention KD in a dynamic feature selection regime, a contribution that we believe is novel.
>
> ---
>
> **Q:** *Can the proposed method generalizes to other object centric representation learning approaches beyond slot attention? I did not find any specific design for the slot attention. I guess other object centric representation learning approaches can also be employed and extended to the federated learning setting. It would be interesting to see these results as well.*
>
> **A:** We agree with the hypothesis that other unsupervised object centric learning like MONET and IODINE will work with our framework, as long as they could be modified to use feature reconstruction instead of image reconstruction.
> In object centric learning literature slot attention has been shown to be most effective in both synthetic toy data and real-word data. We believe slot attention is the strongest candidate to incorporate within our framework, and also representative of the phenomenon of object centric learning with federated learning.
>
> 1. Klaus Greff, et al.,Multi-Object Representation Learning with Iterative Variational Inference (IODINE)
> 2. Christopher P. Burgess, at al., MONet: Unsupervised Scene Decomposition and Representation

---

### Official Review · Reviewer_bGQ6 · 2025-07-02

**Clarity:** 3
**Significance:** 2
**Originality:** 2
**Rating:** 4
**Confidence:** 3

**Summary:**

This paper proposes FORLA, a novel framework for Federated Object-Centric Representation Learning with Slot Attention that couples object-centric structure learning with collaborative feature adaptation across clients. FORLA mitigates conceptual entanglement by preserving local structure—a key challenge in multi-domain learning with semantic overlap. Moreover, while traditional federated learning has primarily focused on classification and segmentation tasks, this work introduces self-supervised object-centric learning into the FL setting, filling an important gap in current research.

**Questions:**

The current design uses a dual-branch architecture, where the student reconstructs raw features and the teacher reconstructs adapted features. However, the motivation behind this specific design choice is not fully explained. A clearer explanation would improve the overall clarity and justification of the method and may positively influence the evaluation.

Federated learning often accelerates the training process due to parallel updates across multiple clients. While the paper reports the overall training time of the federated setup, it is unclear how this compares to centralized training in terms of speedup. Could the authors report the actual speedup ratio relative to centralized training and clarify whether FORLA achieves any acceleration benefits in practice?

In the early stages of training, FORLA updates the teacher adapter using the student via EMA. However, it remains unclear how the update boundary between EMA-based updates and gradient-based updates is defined. Could the authors elaborate on how the two update mechanisms are coordinated and whether any specific schedule or switching criteria is applied?

While FORLA demonstrates strong performance in the reported experiments, real-world data distributions are often more complex and non-stationary. How sensitive is FORLA to different degrees of data heterogeneity? Specifically, would randomly reshuffling the data partitions under varying heterogeneity levels significantly affect its performance?

**Ethical Concerns:**

["NO or VERY MINOR ethics concerns only"]

**Final Justification:**

My concerns have almost been addressed, i will raise my score.

**Limitations:**

yes

**Quality:**

2

**Strengths And Weaknesses:**

Quality.

The paper presents a technically sound and well-executed framework that integrates self-supervised object-centric learning into the FL paradigm. FORLA combines Slot Attention with a dual-branch student-teacher architecture and introduces several thoughtful design choices, such as the use of EMA and cross-branch FedAvg, to enable mutual knowledge transfer and reduce conceptual entanglement. The experimental section is comprehensive, with detailed implementation descriptions and reproducibility considerations. Extensive evaluations across multiple datasets and metrics demonstrate consistent improvements over both individual and centralized baselines. However, some architectural decisions—such as the motivation for dual reconstruction targets and the necessity of cross-branch federation—would benefit from deeper theoretical or empirical justification. In real-world scenarios, federated systems often involve a large number of clients and varying degrees of data heterogeneity. The current experiments are limited in client scale and do not assess the framework’s performance under more complex, realistic distributions, which raises concerns about its practical applicability.

Clarity.

The paper is generally clear and well-organized. A more in-depth explanation of how these components contribute to the mitigation of conceptual entanglement would enhance the overall clarity and persuasiveness of the work.

Significance.

The paper addresses a timely and important challenge in multi-domain learning with overlapping semantics. By introducing object-centric self-supervised learning into the FL context, the authors tackle a largely unexplored area and open up new directions for future research. The demonstrated performance gains across various benchmarks underscore the practical impact of the method.

Originality.

The work is original in both its problem formulation and methodological design. While the components employed are known individually, their integration into a federated setting for object-centric representation learning is novel. The perspective of addressing conceptual entanglement through collaborative object-level learning and cross-client feature adaptation is fresh and contributes meaningfully to the FL literature.

---

> ### Author Rebuttal · Authors · 2025-07-30
>
> We are grateful for the reviewer’s appreciation of the challenge we are addressing and for the acknowledgment that our integration of object-centric learning with federated learning is novel and meaningful for the FL literature. In response, we will elaborate on the dual-branch architecture and cross-branch mechanisms, report training efficiency, and provide additional analysis to address the questions regarding sensitivity to heterogeneity and update strategies. In particular, new experiments with heterogeneous datasets further support our hypothesis that FL lends naturally to unified representation learning across domains by exploiting the local structural coherence present in homogeneous datasets.
>
> ---
>
> **Q:** *The current design uses a dual-branch architecture, where the student reconstructs raw features and the teacher reconstructs adapted features. However, the motivation behind this specific design choice is not fully explained. A clearer explanation would improve the overall clarity and justification of the method and may positively influence the evaluation.*
>
> **A:** Thank you for this comment. Our justifications are:  1) the two branch design allows to learn adaptive features which are useful in delineating objects in complex scenes and novel domains(as shown by the PCA of adapted feature versus raw feature in Fig. 2 ). 2) the reason for one branch to reconstruct raw features while the other reconstructs dynamic features is that reconstructing raw features anchors the learning target which makes convergence possible. Reconstructing dynamic features solely would indeed not converge. We will include this motivation to the manuscript.
>
> ---
>
> **Q:** *Federated learning often accelerates the training process due to parallel updates across multiple clients. While the paper reports the overall training time of the federated setup, it is unclear how this compares to centralized training in terms of speedup. Could the authors report the actual speedup ratio relative to centralized training and clarify whether FORLA achieves any acceleration benefits in practice?*
>
> **A:** Centralized training on 1.4 million data points (frames) across 7 datasets with 130 epochs and batch size 16 takes 21.6 hours which is 1.8 times slower than FL. As the data can not be distributed across too many clients since unsupervised  object centric learning needs a certain amount of data to discover meaningful concept (please also see the answer and results to Reviewer TTbE's comments on data re-partition), it would indeed be interesting to explore in more detail how one can achieve an optimal speedup/performance ratio in a FL training regime.
>
> ---
>
> **Q:** *In the early stages of training, FORLA updates the teacher adapter using the student via EMA. However, it remains unclear how the update boundary between EMA-based updates and gradient-based updates is defined. Could the authors elaborate on how the two update mechanisms are coordinated and whether any specific schedule or switching criteria is applied?*
>
> **A:** We set an empirical switching criteria as 90K iterations, as this number allows the reconstruction loss of slot attention models on clients's student branch to converge to plateau. We also found that training for a larger number of iterations before switching from EMA (larger than 90K) can lead to better initial convergence of the student branch, but in the long term this advantage is offset by the second stage training of the the student, making the additional time spent on initial training obsolete. We will add this detail to our experiments setup. In future, the empirical threshold can be replaced by a dynamic threshold based on tracking of student loss.
>
> ---
>
> **Q:** *While FORLA demonstrates strong performance in the reported experiments, real-world data distributions are often more complex and non-stationary. How sensitive is FORLA to different degrees of data heterogeneity? Specifically, would randomly reshuffling the data partitions under varying heterogeneity levels significantly affect its performance?*
>
> **A:** Thank you for this question. We ran a set of new experiments with data reshuffling and stratification. Based on data stratification in Table 4 of the manuscript, we reshuffle the datasets so that a group of clients (clients count smaller than 7) has random selection from all 7 datasets. We demonstrate experiments under 3 different combinations of datasets. Results for the first two data combinations with four clients show very similar results (when compared to centralized learning) as the initial results on 7 clients. This means that FORLA is quite robust to moderate changes in data heterogeneity. The third data combination used just two clients and here FORLA showed less of an advantage as it could not exploit learning from multiple subdomains.  FORLA's robustness is driven by SA gradually discovering disentangled concepts  across clients.
> A plain mixed centralized training will remove such disentanglement. In centralized training, this problem could be addressed by techniques in supervised learning by leveraging meta data source labels to group training data in batch schedules, but the FL offers an elegant and efficient solution.
>
> | Metric | Method      | Abd (Comb1) | Thor (Comb1) | PASCAL (Comb1) | YTOBJ (Comb1) | Chol (Comb2) | Abd (Comb2) | YTOBJ (Comb2) | YTVIS (Comb2) | Abd (Comb3) | PASCAL (Comb3) |
> |--------|-------------|-------------|--------------|--------------|---------------|--------------|-------------|---------------|---------------|-------------|--------------|
> | mBO    | Individual  | 50.34       | 53.03        | 34.98        | 48.98         | 33.54        | 50.34       | 48.98         | 35.16         | 50.34       | 34.98        |
> |        | Centralized | 53.42       | 50.89        | 35.97        | 45.28         | 31.64        | 54.16       | 47.84         | 37.15         | 56.56       | 36.83        |
> |        | FORLA       | **55.16**   | **57.67**    | **38.27**    | **50.66**     | **34.86**    | **59.63**   | **48.45**     | **38.79**     | **56.84**   | **37.68**    |
> | Corloc | Individual  | 70.73       | 56.62        | 51.10        | 65.11         | 42.36        | 70.73       | 60.31         | 58.08         | 48.98       | 51.10        |
> |        | Centralized | 73.57       | 53.49        | 57.14        | 53.00         | 46.42        | 72.50       | 57.41         | 60.41         | 76.97       | 51.10        |
> |        | FORLA       | **75.52**   | **64.53**    | **57.14**    | **66.44**     | **54.14**    | **77.63**   | **61.74**     | **65.21**     | **79.32**   | **57.69**    |
>
> *Table: mBO and Corloc scores across different methods and data combinations.Abd:Abdominal,Thor:Thoracic,Chol:Cholec, Comb: Data combinations*

---

> > ### Comment · Reviewer_bGQ6 · 2025-08-05
> >
> > Thank you for the response. My concerns have almost been addressed,  i will raise my score.

---

> ### Author Response · Authors · 2025-08-05
>
> Dear Reviewer bGQ6,
>
> We're happy to hear that we’ve managed to address most of your concerns. We appreciate your thoughtful review and thank you very much for raising the score! If there are any outstanding issues we can address, please feel free to let us know.
>
> Best regards,
>
> The Authors

---

### Official Review · Reviewer_TTbE · 2025-07-03

**Clarity:** 2
**Significance:** 3
**Originality:** 3
**Rating:** 5
**Confidence:** 4

**Summary:**

This paper proposes FORLA, a method for federated representation learning with foundation models. The key components of FORLA include a shared adapter to reduce the dimensionality of features extracted from FMs, along with a slot attention mechanism to learn object-centric representations. These parameters are trained in a two-stage distillation process using a student-teacher architecture. Notably, the proposed federated solution exceeds the performance of centralized training on multiple domains.

**Questions:**

1. The ‘Slot FedAvg’ baseline seems quite competitive with FORLA, with only ‘FORLA+AFM’ beating ‘Slot FedAvg + MoE’. Could you provide additional context on the comparison here - why does the student-teacher distillation framework lend itself so well to the AFM adapter?
2. Is this ‘Slot FedAvg’ baseline equivalent to a ‘DINOSAUR FedAvg’ strategy, when we only use features extracted from DINO?
3. The observation that the federated solution can outperform the centralized one is quite interesting. Does this still occur when the number of clients exceeds the number of datasets (and each user correspondingly holds less data)? Further investigation into this phenomenon would be helpful.
4. Have you conducted any ablation studies on the quantity of data available and the optimal feature stack + adapter methods? It would be interesting to see if the AFM still obtains the best results in low-sample settings, and if it is still best to use features from multiple pretrained models.

**Ethical Concerns:**

["NO or VERY MINOR ethics concerns only"]

**Final Justification:**

I appreciate the authors’ efforts during the rebuttal. My concerns regarding the clarity and discussion of the results were addressed, and the additional experiments in more distributed and low-sample settings helped validate the proposed method. In the final version, I encourage the authors to benchmark the full set of baseline methods (e.g., naive Slot-FedAvg) in these added ablations.

**Limitations:**

I like the idea of this paper, but I have some concerns regarding the lack of results in more distributed settings (see weakness #4), and I would like to see a clearer discussion of the key results.

**Paper Formatting Concerns:**

None.

**Quality:**

3

**Strengths And Weaknesses:**

Strengths
1. The paper is well-written and tackles an interesting problem of domain-invariant federated representation learning.
2. The connection of slot attention's success in mixed domains and federated learning is quite interesting.

Weaknesses
1. Some details of the algorithm are difficult to find in the main text and require going into the supplementary material, for example, the description of the two-stage training in Section 3.4.  It would be helpful to bring a more detailed algorithm description to the main text and reserve parts of Section 4 for the appendix instead.
2. The presentation and discussion of the results require some clarification. The authors provide some guiding research questions, but the corresponding discussion of the experiments is difficult to parse. The definition of baseline methods (e.g., Slot FedAvg in Table 2) should also be made more prominent in the text.
3. While the selected baselines make sense, it would be helpful to see a comparison to other recent FL techniques that leverage foundation models, if possible.
4. Some details of the experimental setup are missing and may require some ablation - e.g., the amount of training/validation data on each client. The current results are also limited to 1 client per dataset.

---

> ### Author Rebuttal · Authors · 2025-07-30
>
> We thank the reviewer for recognizing the novelty of connecting slot attention’s success in mixed domains with federated learning, and for appreciating our approach to the timely problem of domain-invariant federated representation learning. We will clarify the algorithmic details in the main text, improve the presentation of baselines, and provide additional ablations and context to address the insightful questions raised. Below is our response to specific questions:
>
> ---
>
> **Q:** *Some details of the algorithm are difficult to find in the main text and require going into the supplementary material, for example, the description of the two-stage training in Section 3.4. It would be helpful to bring a more detailed algorithm description to the main text and reserve parts of Section 4 for the appendix instead.*
>
> **A:** We agree with this proposal to move  details on the two-stage training, including Algorithm 1, from the appendix to the main text. We will move some details on selection of feature adaption modules and parts of datasets description to appendix.
>
> ---
>
> **Q:** *The presentation and discussion of the results require some clarification. The authors provide some guiding research questions, but the corresponding discussion of the experiments is difficult to parse. The definition of baseline methods (e.g., Slot FedAvg in Table 2) should also be made more prominent in the text. Is this 'Slot FedAvg' baseline equivalent to a 'DINOSAUR FedAvg' strategy, when we only use features extracted from DINO?*
>
> **A:** The "Slot FedAvg" baseline differs from the DINOSAUR + FedAvg in that it uses a feature stack and feature adaptation. Because of that, the reconstruction target is the stacked feature. If the baseline had DINO only and a MLP adapter, they would be equivalent. We will improve the description of this and other baselines (currently paragraph starting at line 204) in the revised version.
>
> ---
>
> **Q:** *The 'Slot FedAvg' baseline seems quite competitive with FORLA, with only 'FORLA+AFM' beating 'Slot FedAvg + MoE'. Could you provide additional context on the comparison here - why does the student-teacher distillation framework lend itself so well to the AFM adapter?*
>
> **A:** We thank the reviewer for pointing out this important performance result that we did not elaborate on in the initial submission. While it is true that Slot FedAvg performs well in comparison to FORLA, its performance gains are mostly on Natural datasets while it lags in performance on Surgical datasets. The reason for this is that foundation models are trained on Natural datasets and therefore teacher supervision and feature adaptation are less important for Natural datasets in comparison to Surgical datasets. We think this gap in performance further highlights the flexibility of cross-domain training using FORLA and we'll thus include a stratified average column to Table 2, together with an explanation in the main text.
>
> As for the reason why MOE appears to be a better choice for fedAvg while AFM works better with FORLA's student-teacher distillation: The AFM is much more adaptive and can easily overfit to certain concepts with FedAvg, while with the student-teacher constrained learning in FORLA, AFM's high adaptivity will not hinder scaling to more data.
>
> ---
>
> **Q:** *The observation that the federated solution can outperform the centralized one is quite interesting. Does this still occur when the number of clients exceeds the number of datasets (and each user correspondingly holds less data)? Further investigation into this phenomenon would be helpful.*
>
> **A:** Thank you for this question that made us consider new experiments that show how exploiting structural coherence in local datasets fades when data is distributed. We conducted an additional experiment with 2 clients for each dataset (50% percent of dataset on each) using Abdominal and Thoracic datasets. Indeed, performance is lower in such a setting but federated learning is still able to surpass the centralized learning.
>
> | Region     | Method      | mBO   | HD     | FG-ARI | Cor-Loc |
> |------------|-------------|-------|--------|--------|---------|
> | Abdominal  | Centralized | 54.13 | 41.79  | 60.00  | 74.70   |
> | Abdominal  | FORLA       | 57.30 | 40.07  | 63.30  | 77.08   |
> | Thoracic   | Centralized | 54.52 | 63.82  | 44.37  | 62.67   |
> | Thoracic   | FORLA       | 61.55 | 53.78  | 47.84  | 71.53   |
>
> *Table: Comparison of methods (Centralize and FORLA) across Abdominal and Thoracic using 50% (2 clients each)*
>
> We also performed an experiment with smaller datasets distributed across four clients. First we used all data for each dataset (25% on each client), and then decreased the amount of data to 40% (10% on each client, see also our answer to request for data quantity ablation). Again, we compare Individual (each client learns own model) to Centralized (all data mixed) and FORLA. We can confirm that the advantage of FORLA slowly fades with less data distributed over nodes, indicating that FL might overfit to spurious structural aspects of smaller training batches, highlighting the need for distributed datasets of same domain to have similar distributions.
>
> | Method      | Pascal (25%) |        | Thoracic (25%) |         | Pascal (10%) |        | Thoracic (10%) |         |
> |-------------|--------------|--------|----------------|---------|--------------|--------|----------------|---------|
> |             | mBO          | FG-ARI | mBO            | FG-ARI  | mBO          | FG-ARI | mBO            | FG-ARI  |
> | Individual  | 33.21        | 31.52  | 46.39          | 39.79   | 30.71        | 28.11  | 27.14          | 23.35   |
> | Centralized | 34.98        | **34.16** | 53.03        | 43.48   | **34.99**    | **33.52** | 49.17        | **40.14** |
> | FORLA       | **35.25**    | 34.14  | **54.26**      | **44.30** | 32.09        | 29.25  | **49.66**      | 39.21   |
>
> *Table: Comparison of mBO and FG-ARI across training methods under 25% and 10% data splits for Pascal and Thoracic datasets.*
>
> Since unsupervised  object centric learning needs a certain amount of data to discover meaningful concept, these additional results are consistent with OCL and indicate that our framework requires the data not to be over partitioned.
>
> ---
>
> **Q:** *While the selected baselines make sense, it would be helpful to see a comparison to other recent FL techniques that leverage foundation models, if possible.*
>
> **A:** Federated foundation model research is mostly on LLMs, for instance "Towards Federated Foundation Models: Scalable Dataset Pipelines for Group-Structured Learning". The only vision FL model to our knowledge is "Evaluating Federated DINO's performance on the segmentation task across diverse domains"(Marko Harasic, et al., 2024) which is not object centric and thus could not be quantitatively compared. We will mention this in the discussion/related work.
>
> ---
>
> **Q:** *Some details of the experimental setup are missing and may require some ablation - e.g., the amount of training/validation data on each client. The current results are also limited to 1 client per dataset. Have you conducted any ablation studies on the quantity of data available and the optimal feature stack + adapter methods? It would be interesting to see if the AFM still obtains the best results in low-sample settings, and if it is still best to use features from multiple pretrained models.*
>
> **A:** In addition to our experiment with 50% of  database per client (see answer above), the experiment with 10% of database per client (40% total data) was  designed to partially answer this question. In this setup, FL matched the performance of centralized learning while still outperforming individual training, albeit with diminished returns.
>
> In addition, we previously experimented using only three foundation models (DINO,SAM,CLIP). Both without adaptation (39.78 mBO on abdominal and 42.92 mBO on YTOBJ) and with adaptation (49.58 mBO on abdominal and 47.47 mBO on YTOBJ) results were suboptimal to using four foundation models.
>
> As for the contribution of AFM and feature adaptation in low data regimes, we hypothesize that they would become less stable with increased data heterogeneity and decreased dataset size. We already tested the use of a single foundation model with/without adaptation in the Appendix and the results support our hypothesis. We'll definitely consider a more extensive testing of AFM and other adapters in low data regimes for future work.

---

> > ### Comment · Reviewer_TTbE · 2025-08-08
> >
> > Thanks for your response. For the new results in low-sample and distributed settings, I recommend that the authors add 'Slot FedAvg' as a baseline to more clearly show the benefit of FORLA in the final version. Overall, my concerns about the clarity and discussion of the results were addressed, and I will raise my score.

---

> > > ### Author Response · Authors · 2025-08-08
> > >
> > > Thank you for the thoughtful follow-up. We’re glad we have addressed your concerns, and we appreciate you raising your score! In the final version, we will incorporate the improvements discussed in the rebuttal, and we agree that adding ‘Slot-FedAvg’ baseline for the low-sample and distributed settings can further demonstrate FORLA’s benefits.

---

> ### Author Response · Authors · 2025-08-07
>
> Dear Reviewer TTbE,
>
> Thank you once again for your constructive review and positive feedback on our paper. As the discussion deadline approaches, we wanted to check whether our rebuttal has sufficiently addressed your comments.If you have any remaining concerns, we would appreciate the opportunity to respond before the discussion period ends.
>
> Thank you again for your valuable feedback.
>
> Best regards,
>
> Authors

---

### Decision · Program_Chairs · 2025-09-17

**Decision:**

Accept (poster)

**Comment:**

The paper adapts the Slot Attention-based Object-Centric Representation Learning method to federated settings. In addition, it incorporates a two-stage distillation process and an adapter mechanism into the proposed solution.

Strengths: The paper extends federated learning research into a new application scenario: object-centric representation learning.

Weaknesses: The overall solution is presented in a relatively engineering-oriented manner, with limited theoretical depth.

Reviewer Consensus: All reviewers provided positive scores for this paper.

Recommendation: I recommend accepting this paper, as it opens a promising new sub-domain for federated learning research.